# Deep Neural Cellular Potts Models

**Koen Minartz** [1]  **Tim d'Hondt** [1]  **Leon Hillmann** [2 1 3]  **Jörn Starruß** [4]  **Lutz Brusch** [4]  **Vlado Menkovski** [1]

## Abstract

The cellular Potts model (CPM) is a powerful computational method for simulating collective spatiotemporal dynamics of biological cells. To drive the dynamics, CPMs rely on physics-inspired Hamiltonians. However, as first principles remain elusive in biology, these Hamiltonians only approximate the full complexity of real multicellular systems. To address this limitation, we propose NeuralCPM, a more expressive cellular Potts model that can be trained directly on observational data. At the core of NeuralCPM lies the *Neural Hamiltonian*, a neural network architecture that respects universal symmetries in collective cellular dynamics. Moreover, this approach enables seamless integration of domain knowledge by combining known biological mechanisms and the expressive Neural Hamiltonian into a hybrid model. Our evaluation with synthetic and real-world multicellular systems demonstrates that NeuralCPM is able to model cellular dynamics that cannot be accounted for by traditional analytical Hamiltonians.

## 1. Introduction

Cell migration and multicellular self-organization are crucial biological processes that drive many phenomena of life, such as embryo growth and the spread of cancer (Friedl & Gilmour, 2009; Gottheil et al., 2023). Understanding these cellular dynamics is not only a fundamental goal of

[1]Department of Mathematics and Computer Science, Eindhoven University of Technology, Eindhoven, the Netherlands [2]Department of Applied Physics and Science Education, Eindhoven University of Technology, Eindhoven, the Netherlands [3]Department of Biomedical Engineering, Eindhoven University of Technology, Eindhoven, the Netherlands [4]Center for Information Services and High Performance Computing, TUD Dresden University of Technology, Dresden, Germany. Correspondence to: Koen Minartz <k.minartz@tue.nl>, Lutz Brusch <lutz.brusch@tu-dresden.de>, Vlado Menkovski <v.menkovski@tue.nl>.

*Proceedings of the 42ⁿᵈ International Conference on Machine Learning*, Vancouver, Canada. PMLR 267, 2025. Copyright 2025 by the author(s).

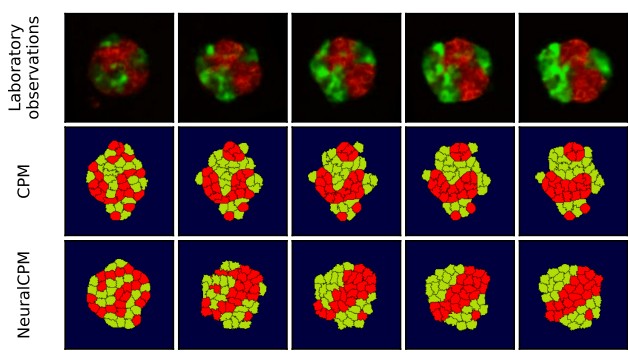

*Figure 1.* NeuralCPM is more expressive than traditional cellular Potts models, allowing for more accurate simulation of real-world collective cell dynamics, e.g. bi-polar self-organization of biological cells over 12 hours in the top row (Toda et al., 2018).

biology, but also needed for the development of medical treatments. As experimental data alone do not reveal the regulatory logic and self-organization principles of complex cell-cell interactions, computational models have to be combined with biological experiments (Marée & Hogeweg, 2001; Hester et al., 2011; Boutillon et al., 2022). One of the most powerful and widely-used numerical methods is the cellular Potts model (CPM), which captures the stochastic movement and shape of cells, interactions in multicellular systems, and multiscale dynamics (Graner & Glazier, 1992; Balter et al., 2007).

CPMs are based on a *Hamiltonian* or *energy function* which maps each possible state in the discrete state space of a multicellular system to a scalar (the energy). To model the evolution of the system over time, a CPM-based simulator stochastically perturbs the current state towards states with a lower energy, following the principles of statistical mechanics. Consequently, the Hamiltonian directly drives the simulated dynamics of cells, and designing the Hamiltonian is the core challenge to arrive at realistic CPM simulations. So far, domain experts need to engineer a tailor-made Hamiltonian for each new problem setting. This Hamiltonian generally consists of a weighted sum of symbolic, physics-inspired features of the system, but (1) it is labor-intensive to develop and (2) arguably only partially captures the full complexity of cellular systems due to simplifying assumptions in the structure of the Hamiltonian.

In this work, we tackle these two weaknesses by presenting *NeuralCPM*: a method for learning cellular Potts models with expressive neural network-based Hamiltonians. In contrast to current practice in the field, NeuralCPM enables fitting a Hamiltonian directly on observational data, without requiring any assumptions on the structure of the Hamiltonian or problem-specific feature engineering. NeuralCPM also facilitates the seamless integration of biological domain knowledge by using the neural network as a *closure term*, complementary to an analytical Hamiltonian based on domain knowledge. This allows us to constrain the learned model to cellular configurations with guaranteed biological realism (e.g. compact cells of given number even for unseen tasks, as opposed to potential hallucinations of fragmented or supernumerous cells), while leveraging the neural network to find structure in the observed data that is too complex to model with an analytical Hamiltonian.

Our main contributions are summarized as follows:

- We propose the neural cellular Potts model, a CPM in which the Hamiltonian is parameterized with a novel neural network architecture that respects the symmetries that are universal in cellular dynamics modeling. We exploit the strong connection between CPMs and deep energy-based models, a generative modeling framework developed in the machine learning community, to directly train the Neural Hamiltonian on observational data.

- We show how known biological mechanisms can straightforwardly be integrated in the NeuralCPM framework by using the Neural Hamiltonian as a closure model. We find that such *biology-informed* Neural Hamiltonians not only improve biological consistency of the simulations, but also act as a regularizer that effectively stabilizes the training process, which can be notoriously challenging for deep energy-based models.

- We validate the effectiveness of our proposed method on three experimental scenarios: (1) parameter fitting of a known analytical cellular Potts model (validation of the learning algorithm); (2) fitting Hamiltonians that are difficult or impossible to attain with analytical functions (validation of the increased expressiveness); and (3) fitting a Hamiltonian on real-world biological data (demonstration of an application to real-world problems).

## 2. Background and Related Work

### 2.1. Energy-Based Models

Energy-based models (EBMs) specify a probability distribution over a random variable $x$ defined on a space $\mathcal{X}$ up to

an unknown normalization constant as follows:

$$p_\theta(x) = \frac{e^{-H_\theta(x)}}{Z_\theta}, \qquad (1)$$

where $H_\theta : \mathcal{X} \to \mathbb{R}$ defines a scalar-valued *energy function* (also called Hamiltonian) parameterized by $\theta$. $Z_\theta = \int_x e^{-H_\theta(x)} dx$ defines the typically intractable normalization constant (LeCun et al., 2006; Song & Kingma, 2021). As $Z_\theta$ does not need to be computed, $H_\theta(x)$ can be any nonlinear regression function as long as it could in principle be normalized (Song & Kingma, 2021), making EBMs a highly flexible class of models. If $H_\theta(x)$ is a neural network, we call the model a *Deep EBM*.

Typically, Deep EBMs are fitted to a target distribution $p^*(x)$ by minimizing the negative log-likelihood:

$$\min_\theta \mathcal{L}(\theta) = \mathbb{E}_{x \sim p^*(x)}[-\log p_\theta(x)], \qquad (2)$$

for which gradient descent on $\theta$ is the de-facto optimization algorithm. The gradient of $\mathcal{L}(\theta)$ then looks as follows (Hinton, 2002; Song & Kingma, 2021):

$$\nabla_\theta \mathcal{L}(\theta) = \mathbb{E}_{p^*(x)}[\nabla_\theta H_\theta(x)] - \mathbb{E}_{p_\theta(x)}[\nabla_\theta H_\theta(x)], \qquad (3)$$

where the expectations can be estimated with Monte Carlo sampling. The main challenge lies in sampling $x$ from the intractable distribution $p_\theta(x)$ to estimate $\mathbb{E}_{p_\theta(x)}[\nabla_\theta H_\theta(x)]$, which is typically achieved by a Markov Chain Monte Carlo (MCMC) algorithm.

### 2.2. Cellular Potts Model

The CPM is a stochastic numerical method used for the simulation of individual and collective cell dynamics (Graner & Glazier, 1992; Savill & Hogeweg, 1997; Balter et al., 2007). In the CPM, cells are modeled as discrete entities with explicit two- or three-dimensional shapes. Because of its capabilities in modeling collective cell behavior, stochastic cellular dynamics, and multiscale phenomena, the CPM has become one of the most effective frameworks for simulating multicellular dynamics (Rens & Edelstein-Keshet, 2019; Hirashima et al., 2017).

The CPM works as follows: given a lattice $L$ and a set of cells $C$, $x^t \in C^{|L|}$ denotes the time-varying state of a multicellular system, where $x_i^t = c$ if cell $c \in C$ occupies the lattice site $i \in L$. The state $x^t$ is evolved over time by an MCMC algorithm that has a stationary distribution characterized by a Hamiltonian $H : C^{|L|} \to \mathbb{R}$. The MCMC dynamics mimic the protruding dynamics of biological cells: a random lattice site $i$ is chosen and its state $x_i$ is attempted to alter to the state of a neighboring site $j$. The proposed state transition $x \to x'$ is accepted with probability $\min\{1, e^{-\Delta H/T}\}$, where $\Delta H = H(x') - H(x)$ is the difference in energy between the proposed state $x'$

and current state $x$, and $T$ is the so-called *temperature* parameter.

As the Hamiltonian $H$ determines the stationary distribution of the Markov chain, the design of $H$ is the key challenge to achieve realistic simulations of cellular dynamics with the CPM. Generally, $H$ contains contact energy and volume constraint terms, as originally proposed in (Graner & Glazier, 1992), and additional application-specific components:

$$
H(x) = \underbrace{\sum_{i,j \in \mathcal{N}(L)} J\left(x_i, x_j\right)\left(1 - \delta_{x_i, x_j}\right)}_{\text{contact energy}}
$$
$$
+ \underbrace{\sum_{c \in C} \lambda\left(V(c) - V^*(c)\right)^2}_{\text{volume constraint}} + H_{\text{case-specific}}(x). \quad (4)
$$

In Equation 4, $\mathcal{N}(L)$ denotes the set of all pairs of neighboring lattice sites in $L$, $J\left(x_i, x_j\right)$ is a contact energy defining adhesion strength between cells $x_i$ at site $i$ and $x_j$ at site $j$, and $\delta_{x,y}$ is the Kronecker delta. Furthermore, $V(c)$ is the volume of cell $c$, $V^*(c)$ is $c$'s target volume, and $\lambda$ is a Lagrange multiplier. Since the first introduction of the CPM, many extensions for $H_{\text{case-specific}}$ have been proposed for varying biological applications, taking into account external forcing, active non-equilibrium processes like chemotaxis, and many other biological concepts (Hirashima et al., 2017). However, as opposed to physical systems like evolving foams, where the Hamiltonian can be derived from first principles and which have been modeled with the CPM (Graner et al., 2000), an equivalent of first principles remains elusive for living systems. Therefore, the task of designing a suitable Hamiltonian requires significant domain expertise and needs to be repeated for each new case. Moreover, even well-designed Hamiltonians will only partially account for the observed cell dynamics.

## 2.3. Neural Networks for Cellular Dynamics Simulation

Neural networks have gained traction as simulation models of complex dynamical systems. The primary objective of most works in this field has been to improve computational efficiency over physics-based numerical simulators (Azizzadenesheli et al., 2024; Li et al., 2021; Kochkov et al., 2021; Gupta & Brandstetter, 2023). However, models of multicellular systems including vertex-based models, phase-field models, and the CPM generally only partially explain the dynamics observed in laboratory experiments (Alert & Trepat, 2020; Brückner & Broedersz, 2024). Here the value of neural simulators lies primarily in *improved accuracy* and *new discoveries*, rather than accelerated simulation.

Still, little research has been done in machine learning-driven modeling of cellular dynamics, and as opposed to

this work, most methods consider cells as point masses without an explicit shape (LaChance et al., 2022; Yang et al., 2024). Although some machine learning methods for modeling CPM-like dynamics of single-cell (Minartz et al., 2022) and multicellular systems (Minartz et al., 2023) have been proposed, these are autoregressive models that require sequence data covering full trajectories of cellular dynamics for training, which can be costly to acquire. Moreover, these methods are black-box surrogates, and cannot exploit any biological knowledge about the system. In contrast, Neural-CPM requires only observations of self-organized states as training data, akin to Neural Cellular Automata (Mordvintsev et al., 2020), and relies on the powerful CPM framework to reconstruct the dynamics of cells.

## 3. Neural Cellular Potts Models

### 3.1. Neural Hamiltonian Architecture

Our goal is to learn a Neural Hamiltonian $H_\theta$ that parameterizes a CPM of which the stochastic dynamics align with the complex distribution over real cellular behavior. A challenge in modeling physical systems is that they often admit multiple equivalent representations, a property formalized through mathematical symmetries. Incorporating these symmetries in neural network architectures can greatly enhance model performance (Bronstein et al., 2021). In the context of CPMs, symmetries arise in Hamiltonians that are *invariant* to permutations of the set of cells $C$ and translations of the lattice $L$. As such, for any transformation $g$ that permutes $C$ and translates $L$, the Neural Hamiltonian should satisfy $H_\theta(x) = H_\theta(gx)$.

Generally, GNNs (Gilmer et al., 2017; Kipf & Welling, 2017), Deep Set models (Zaheer et al., 2017), or transformers (Vaswani et al., 2017) would be the designated building blocks for architectures that respect permutation symmetry. However, in the CPM context, such architectures do not apply, as they operate on node features represented by vectors, whereas in our case $x$ represents a regular lattice. Instead, we propose a Neural Hamiltonian architecture that is invariant to both translations and permutations, illustrated in Figure 2. The global structure of this architecture is as follows: first, $x$ is one-hot encoded, such that for each cell $c \in C$, we now have a separate grid $h_c^0$ representing that cell. $(h_c^0)_i$ equals 1 if $c$ occupies lattice site $i$, and 0 otherwise. Then, we iteratively process the embeddings $\left\{h_c^l\right\}_{c=0}^{|C|}$ with the $l$'th NH layer to produce a deep equivariant representation of the system. Each NH layer processes their input cell embeddings by passing each cell's embedding independently through a neural network $\phi^l$. $\phi^l$'s outputs $h_c'$ are then aggregated using a permutation-invariant function $\bigoplus$, in our case summation, to yield a global context lattice $A$ of the entire system. Finally, all $h_c^l$ are processed in tandem with $A$ by the cell-interaction network $\psi^l$, after which local

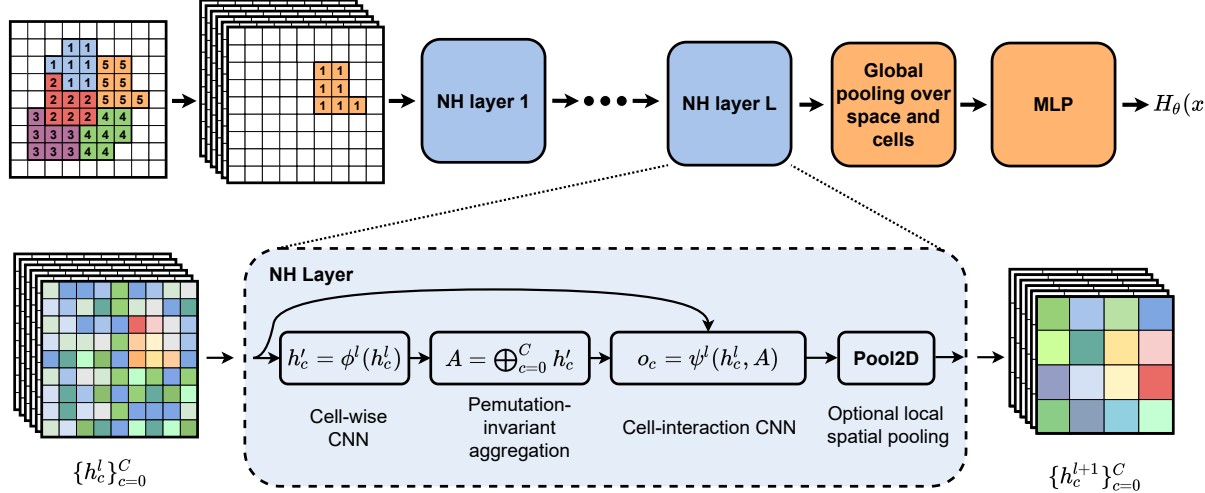

*Figure 2.* Architecture of the Neural Hamiltonian (NH). First, the discrete CPM input undergoes a pixel-wise one-hot encoding. Then, $L$ iterations of NH layers are applied to extract a deep representation of the system that is equivariant to translations and permutations of cell indices. Finally, the extracted representation is pooled globally over the spatial and cell axes, yielding an invariant global representation of the system which is processed by a multi-layer perceptron to compute the Hamiltonian value.

spatial pooling, for example max-pooling, can be applied to compress the representation of the system. To respect the translation symmetry of the problem and promote localized pattern recognition, we parameterize $\phi^l$ and $\psi^l$ with convolutional neural networks (CNNs). After processing by the NH layers, we pool the embeddings to obtain an invariant representation before processing them with a shallow multi-layer perceptron to compute the Hamiltonian $H_\theta(x)$.

**Biology-informed Hamiltonians.** A key advantage of NeuralCPM is that it closely follows the cellular Potts modeling paradigm, which enables us to seamlessly integrate biological domain knowledge. Specifically, even though symbolic Hamiltonians are approximations, they may still account for partially known mechanisms underlying the observed dynamics. In this case, we wish to exploit this domain knowledge to expedite the model fitting task and to yield a more interpretable model. We achieve this by using the Neural Hamiltonian as a *closure model* on top of an interpretable symbolic Hamiltonian:

$$H_\theta(x) = w_S \cdot H_{\theta^S}(x) + w_{NN} \cdot H_{\theta^{NN}}(x), \quad (5)$$

where $H_{\theta^S}(x)$ is the symbolic component and $H_{\theta^{NN}}(x)$ is the Neural Hamiltonian component, and $w_S$ and $w_{NN}$ are learned weights to balance the contributions of both components. If the parameters $\theta^S$ of the symbolic component are not known, $H_\theta(x)$ can be trained end-to-end as long as $H_{\theta^S}(x)$ is differentiable with respect to $\theta^S$. Through its biology-informed structure, $H_{\theta^S}(x)$ expresses a strong prior on the overall Hamiltonian $H_\theta$, while $H_{\theta^{NN}}(x)$ is responsible for expressing higher-order terms that cannot be

accounted for by $H_{\theta^S}(x)$. In this work, we consider the volume constraint and the contact energies of Equation 4 as symbolic components; depending on the available knowledge of the biological mechanisms in the system at hand, more sophisticated expressions can be included.

### 3.2. Training

Our training strategy is to minimize the negative log-likelihood objective (Equation 2) using gradient descent (Equation 3). Given a dataset $\mathcal{D}$ of observed cellular systems, we estimate the expectation over $p^*(x)$ in Equation 3 with a batch of $B$ datapoints $\{x_b^+\}_{b=1}^B$, sampled uniformly from $\mathcal{D}$. The expectation over $p_\theta(x)$ in Equation 3 is estimated with a batch of samples $\{x_b^-\}_{b=1}^B$ obtained using $B$ independent MCMC chains. Inspired by (Du & Mordatch, 2019), we also add a regularization term weighted by a small scalar $\lambda$ to the objective to regularize energy magnitudes of $H_\theta(x)$ and improve numerical stability. This gives the loss estimate:

$$\hat{\mathcal{L}}(\theta) = \frac{1}{B} \sum_{b=1}^B H_\theta(x_b^+) - H_\theta(x_b^-)$$
$$+ \lambda \left( H_\theta(x_b^+)^2 + H_\theta(x_b^-)^2 \right) \quad (6)$$

The computational complexity of training is dominated by running the MCMC sampler to obtain samples from $p_\theta(x)$. Each MCMC step requires a forward pass of $H_\theta(x)$ and typically many thousands of MCMC steps are required before the chain has converged to an equilibrium sample of $p_\theta(x)$. Therefore, we introduce an approximate sampler that accelerates the original CPM sampler by performing multiple

spin-flip attempts in parallel. In principle, this could lead to synchronization artifacts as parallel spin-flips might affect each other (Sultan et al., 2023); however, we stress that this approach should be regarded as a surrogate sampling procedure primarily intended to accelerate training, analogous to how well-established EBM training techniques also successfully trade-off unbiased sampling for faster training (Tieleman, 2008). More details about the training algorithm, the approximate sampler and implementation can be found in Appendix B.

## 4. Experiments

### 4.1. Experiment Setup

**Objectives and scope.** Our experiments aim to: (1) validate the effectiveness of the NeuralCPM training algorithm for fitting Hamiltonians to data, and (2) evaluate the capability of using a Neural Hamiltonian to model cellular dynamics and self-organization in synthetic and real-world scenarios. To the best of our knowledge, no prior methods have been developed for learning an energy function to model cellular dynamics. Hence, we compare our approach to alternative neural network architectures for the Hamiltonian as well as analytical CPMs. Details on experiments and datasets are in Appendix A and in our code.[1]

**Experimental scenarios and datasets.** To address objective (1), we generate synthetic datasets using the cell-sorting Hamiltonian of Equation 4 (Graner & Glazier, 1992), and measure how well the training algorithm can learn the parameters in this analytical function from data. Different parameterizations lead to different dynamics, and we generate data following the type A, B, D and F regimes as illustrated in (Edelstein-Keshet & Xiao, 2023) and Figure 3.

To address objective (2), we consider two experimental scenarios. For the first scenario, we introduce the *Cellular MNIST* dataset: a synthetic dataset in which cells form digit-like structures, also illustrated in Figure 3. The goal of this scenario is not to model cellular structures that are biologically relevant, but rather to investigate a well-controlled case that is too complex to model with an analytical Hamiltonian and simultaneously allows for relatively straightforward visual verification of simulation quality.

The second scenario for objective (2) concerns a biological experiment by Toda et al. (2018). A hallmark during embryo development is the self-organization of the principal body axis from an unstructured group of cells, which can be recapitulated with in-vitro experiments and quantified using time-lapse microscopy (Toda et al., 2018). Here, we choose the observation of a bipolar axis formation in cell aggregates as shown in Figure 1, and we refer to this scenario as *bipolar*

[1]https://github.com/kminartz/NeuralCPM

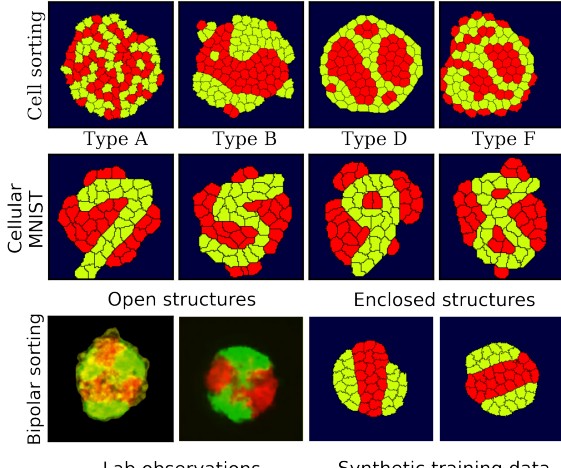

*Figure 3.* Data used in the experiments. Top row: example datapoints of different cell sorting regimes, as illustrated in Edelstein-Keshet & Xiao (2023). Middle row: example datapoints from the Cellular MNIST dataset. Bottom row: example datapoints of Toda et al. (2018) (leftmost two images) and synthetic counterparts (rightmost two images). The synthetic counterparts are used for training, after which we validate the model against the real-world data of Toda et al. (2018).

*axial organization*. This behavior is surprising because the cells of two different types, expressing (after induction) different type-specific P-cadherin or N-cadherin adhesion molecules, were expected to sort into a concentric or unipolar configuration (Graner & Glazier, 1992).

As Toda et al. (2018) performed only six repetitions of this experiment, we construct synthetic counterparts of the final configurations for training using Morpheus (Starruß et al., 2014). Specifically, we ensure that cells from the 'red' type move into a central band from random initial conditions, and that the other cells move towards two opposing clusters along the horizontal axis. To achieve this, we prescribe a preferred target coordinate for each cell independently by adding an additional term to the Hamiltonian that penalizes distance from the target coordinate. Additionally, to make sure the training data are isotropic, we randomly rotate the generated configuration. In contrast to this synthetic data generation procedure, biological cells (and NeuralCPM) do not rely on any such prescribed target locations, but update cell positions as a result of cell-cell interactions in the current configuration. Of course, such a synthetic training data approach should be taken with care. Consequently, we validate the cellular dynamics predicted by the trained NeuralCPM model against the real biological dynamics reported in the Supplemental Figure S6B of Toda et al. (2018) to assess model quality.

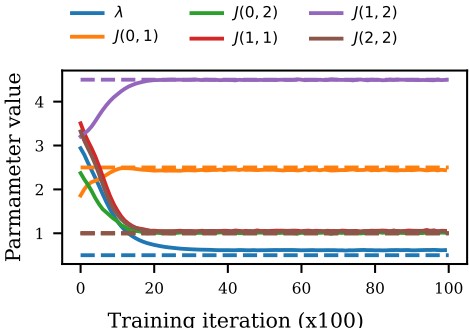

*Figure 4.* Convergence of the parameters for Type B cell sorting. Dashed lines indicate the true values, solid lines indicate the learned values ($T = T^*$) over the course of training.

*Table 1.* Error of fitted coefficients for different parameterizations of the cell sorting Hamiltonian.

| Cell sorting regime (Edelstein-Keshet & Xiao, 2023) | A | B | D | F |
|---|---|---|---|---|
| $\log_{10}(\text{RMSE}) - T = 1$ | -1.26 | -0.01 | -1.08 | -0.99 |
| $\log_{10}(\text{RMSE}) - T = T^*$ | -1.67 | -0.75 | -1.18 | -1.20 |

### 4.2. Fitting Analytical Hamiltonians

**Metrics and baselines.** To assess how well our learning algorithm can fit analytical Hamiltonians to data, we measure the Root Mean Squared Error (RMSE) of the learned parameters of the Hamiltonian. Since the temperature parameter is mainly related to fluctuations in the system over time and not so much to the statistical properties of the system at equilibrium, it is poorly identifiable from static snapshots. Hence, we report the RMSE for both temperatures $T = 1$ and $T = T^*$, where $T^*$ is the temperature that minimizes the RMSE between the learned coefficients and the ground truth.

**Results.** As shown in Table 1, the final learned parameters approximate the ground-truth very well. In addition, as can be seen in Figure 4, the parameters converge rapidly to the true values, highlighting the efficiency of the learning algorithm. Additional results illustrating the learning process and the robustness of the parameter fitting can be found in Appendix C.1.

### 4.3. Cellular MNIST

**Metrics and baselines.** To assess the simulated dynamics, we consider both the cell and collective perspectives. From an individual cell point of view, we require cells to have a realistic volume and to be contiguous. Consequently, we measure $p_{\text{volume}}$, the proportion of states in which all cells exceed the lowest and highest measured cell volumes in the training data by at most 10% of the mean cell volume, and

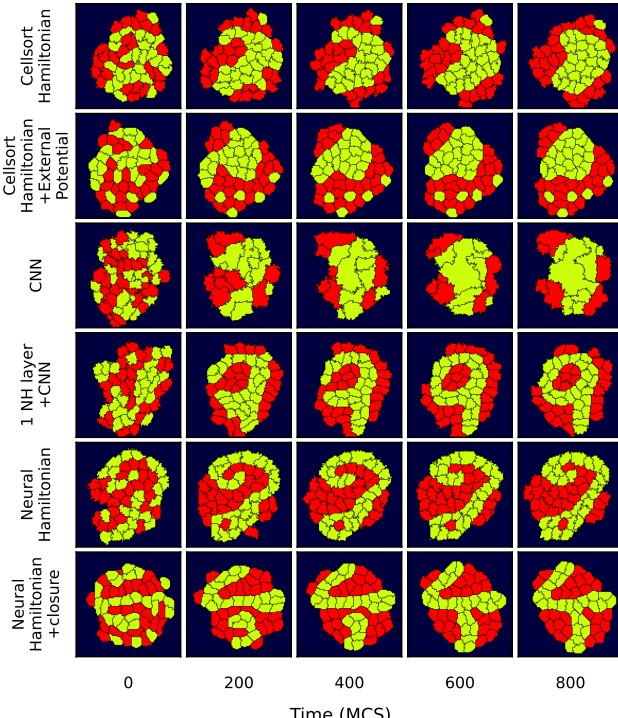

*Figure 5.* Qualitative results for dynamics simulated by CPMs with varying Hamiltonian models trained on Cellular MNIST data.

$p_{\text{unfragmented}}$, the proportion of states in which there are at most three fragmented cells. We allow for three fragmented cells as CPMs allow for small temporary fragmentations. At the collective scale, our goal is to assess whether cells successfully organize into digit-like structures. Following the Inception Score (Salimans et al., 2016), a metric used to quantify the quality of generative models for natural images, we calculate a *Classifier Score* (CS) using a classifier $p_\phi(y|x)$ that we trained on the cellular MNIST dataset:

$$CS = \exp\left(\mathbb{E}_{x \sim p_\theta(x)}\left[KL\right]\right),$$
$$KL = D_{KL}\left(p_\phi(y|x) || \mathbb{E}_{x' \sim p_\theta(x')}\left[p_\phi(y|x')\right]\right). \quad (7)$$

Although $CS$ is calculated in the same way as the established Inception Score, the classifier $p_\phi$ is not as well-validated as the Inception network. Consequently, the metric should be interpreted as a rough indication of quality complementary to visual inspection, where high $CS$ indicates distinct and diverse cellular structures.

As baselines, we compare against two analytical models as well as neural network-based Hamiltonians. The analytical models are the prototypical cell sorting Hamiltonian (Graner & Glazier, 1992) (Equation 4) and the cell sorting Hamiltonian plus a learnable external potential. The goal of the neural network baselines is to evaluate the design choices in the Neural Hamiltonian. To this end, we consider (1) a Convolutional Neural Network; (2) a 1-layer Neural Hamil-

*Table 2.* Results on Cellular MNIST data. $p_{volume}$ and $p_{unfragmented}$ assess the validity of the dynamics at the cell level from a biological perspective, while CS assesses to what extent cells successfully assemble in distinct digit-like structures.

| Model | $p_{volume} \uparrow$ | $p_{unfragmented} \uparrow$ | CS $\uparrow$ |
|---|---|---|---|
| Cellsort Hamiltonian | **1.00** | **1.00** | 2.47 |
| Cellsort + External Potential | **1.00** | **1.00** | 3.11 |
| CNN | 0.00 | 0.05 | 3.70 |
| 1 NH layer + CNN | 0.06 | 0.87 | 3.53 |
| Neural Hamiltonian | 0.11 | 0.99 | **4.91** |
| Neural Hamiltonian + closure | **1.00** | **1.00** | 4.35 |

tonian, followed by invariant pooling over representations of cells and a CNN; (3) the vanilla Neural Hamiltonian as illustrated in Figure 2; and (4) a Neural Hamiltonian as closure on top of a cell sorting Hamiltonian with learnable parameters. Models (1) and (2) serve to investigate the importance of deep equivariant embeddings over architectures relying on representations without permutation symmetry (1) or on invariant representations (2), while comparing (3) and (4) gives insight on the relevance of including biological knowledge in the Hamiltonian. Details on the model architectures can be found in Appendix B.2. Additional results concerning further Hamiltonian baselines and ablations can be found in Appendix C.2.

**Results.** Figure 5 shows qualitative results of simulated trajectories starting from a mixed cell cluster configuration; more visualizations can be found in Appendix C.3. As expected, the analytical models are not able to capture complex non-linear relationships, reflected in the failure of cells to form a digit-like structure. The CNN Hamiltonian produces dynamics that are clearly unrealistic, because it lacks the inductive bias of permutation symmetry. In contrast, the architectures based on Neural Hamiltonians respect the symmetries of the system and lead to cells organizing in digit-like structures.

In line with these qualitative results, Table 2 shows that the analytical models excel in the biological metrics $p_{volume}$ and $p_{unfragmented}$ due to the biology-informed design of their Hamiltonians, but achieve low $CS$ values as they are not expressive enough to model digit-like structures. From the Neural Hamiltonian models, NH achieves the highest CS score, which is substantially higher than the 1 NH layer + CNN model, stressing the relevance of deep equivariant representations. However, these models are subject to unsatisfactory biological metrics. In contrast, using the NH as a closure term yields high scores on the biological and $CS$ metrics, as it enjoys the strong biological structure of the analytical component to constrain the dynamics to be biologically realistic, while using the more expressive NH architecture to guide cells towards digit-like formations.

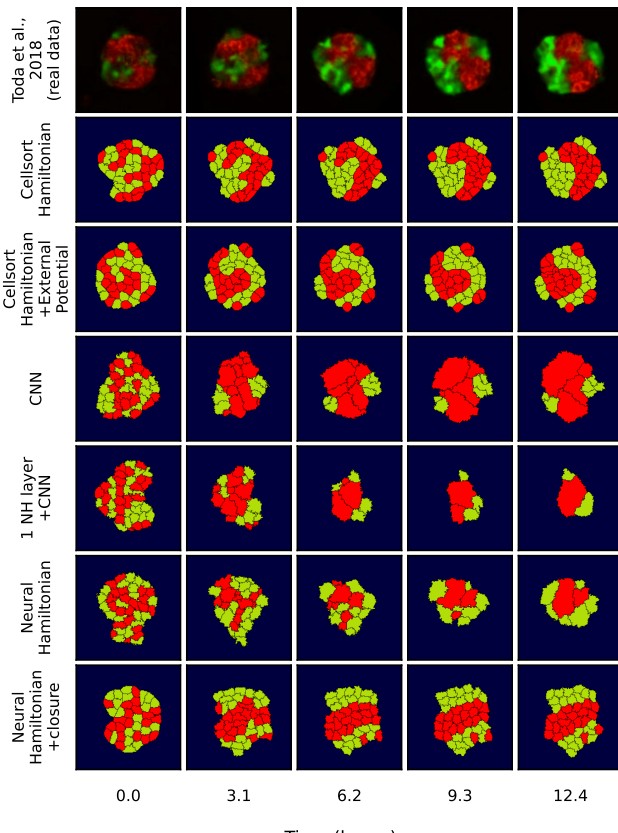

*Figure 6.* Biological dynamics observed in Toda et al. (2018), and qualitative results for dynamics simulated by CPMs with varying Hamiltonian models trained on bipolar axial organization data.

### 4.4. Bipolar Axial Organization

**Metrics and baselines.** As in Section 4.3, we evaluate simulations for biological consistency and collective behavior. For the biological metrics, we use the same indicators $p_{volume}$ and $p_{unfragmented}$. For the collective dynamics, we quantify the bipolar axial organization as follows: first, we rotate the image along the principal axis of the polar cell clusters (green in Figure 3). We then measure the variance of the spatial configuration of each cell type along this axis as well as along the orthogonal axis. We consider the same baselines as in Section 4.3, and provide additional results in Appendix C.2.

**Results.** Figure 6 shows a microscopy time-lapse by Toda et al. (2018), as well as CPM-simulated trajectories for different Hamiltonians; more simulated trajectories can be found in Appendix C.3. As in Section 4.3, the analytical baselines fail to capture bipolar structures due to their insufficient expressiveness, and the CNN-Hamiltonian produces distorted dynamics. However, in this case the Neural Hamiltonian-based models without closure term fail to pro-

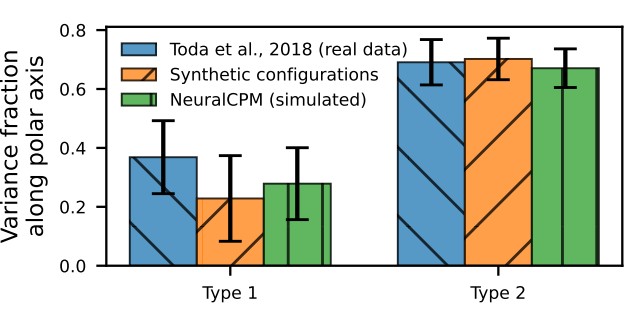

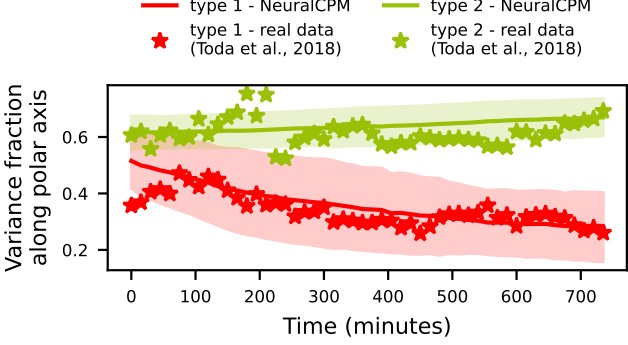

(a) Bipolar cellular organization at equilibrium.

(b) Development of bipolar cellular organization over time.

*Figure 7.* Bipolar cellular organization as quantified through the fraction of variance along the polar axis for each cell type. A high value indicates strong alignment with the bipolar axis, which is expected for type 2 cells in equilibrium, whereas a low value indicates alignment in the orthogonal direction. Almost all observations of Toda et al. (2018) lie within 1 standard deviation of the mean of the simulations (error bars in (a) and shaded in (b)), indicating that NeuralCPM reproduces the observed bipolar organization behavior.

*Table 3.* Results on bipolar axial organization for CPMs with different Hamiltonians. We use the same biological consistency indicators $p_{\text{volume}}$ and $p_{\text{unfragmented}}$ as in Table 2, as well as the RMSE of the variance along the polar and orthogonal axes of the two cell types to quantify how well bipolar axial organization is captured.

| Model | $p_{\text{volume}} \uparrow$ | $p_{\text{unfragmented}} \uparrow$ | Axial alignment ↓ RMSE |
|---|---|---|---|
| Cellsort Hamiltonian | **1.00** | **1.00** | 147.2 |
| Cellsort + External Potential | **1.00** | **1.00** | 154.4 |
| CNN | 0.00 | 0.07 | 153.5 |
| 1 NH layer + CNN | 0.00 | 0.11 | 329.1 |
| Neural Hamiltonian | 0.00 | 0.17 | 254.2 |
| Neural Hamiltonian + closure | 0.77 | **1.00** | **37.3** |

duce reasonable dynamics, due to fast divergence of these models during training, a common issue of EBMs. In our Cellular MNIST experiment, we observed a similar tendency, but we were able to mitigate divergence by careful hyperparameter tuning, which we were not able to achieve for the bipolar axial sorting. In contrast, the NH+closure model trained stably out of the box, with minimal adaptations compared to the Cellular MNIST design. As such, we empirically observe that the biologically informed analytical term in NH+closure not only improves the biological realism of the simulations, as evident from Table 3, but also acts as an effective regularizer that stabilizes training.

To compare the NH + closure model with the laboratory observations of Toda et al. (2018), we use the six recorded self-organized states as well as a time-lapse of the self-organizing process, which the authors kindly shared with us. Figure 7(a) shows the degree of bipolar organization in the observations, our synthetic training data, and the Neural-CPM simulations. Due to their synthetic nature, our training data do not contain as much of the noise that is inherent to real observations, which is expressed in the slightly more organized configurations. Our NeuralCPM simulations yield self-organization patterns that largely overlap with the observations of Toda et al. (2018). Moreover, Figure 7(b) shows that the self-organizing dynamics in our simulations align with the time-lapse of Toda et al. (2018). Crucially, while the cells in the synthetic training data could only be prepared in a predefined final bipolar configuration, NeuralCPM successfully predicts the temporal dynamics and spontaneous symmetry breaking observed in the experiments by Toda et al. (2018). A Neural Hamiltonian trained on multicellular structures paired with the well-established CPM dynamics can therefore not only be used to study equilibrium configurations, but also elucidates the dynamic pathways of cellular self-assembly towards such states.

## 5. Conclusion

This work introduced NeuralCPM, a method for simulating cellular dynamics with neural networks. Whereas the current practice in CPM research is for domain experts to hand-craft an approximate symbolic Hamiltonian for each application, NeuralCPM parameterizes the Hamiltonian with an expressive neural network. These Neural Hamiltonians can model more complex dynamics than analytical Hamiltonians and can be trained directly on observational data. Our results demonstrated that incorporating the symmetries of multicellular systems is crucial to train an effective Neural Hamiltonian. Moreover, we found that using the Neural Hamiltonian as a closure term on top of a biology-informed symbolic model improves biological realism and training stability. Finally, a case study on real-world complex self-organizing dynamics showed that NeuralCPM's simulations successfully predict laboratory observations. We conclude

that NeuralCPM can effectively model collective cell dynamics, enabling the study of more complex cell behavior through computer simulations.

**Limitations and future work.** As the aim of this work was to introduce and validate the core concepts of Neural-CPM, our evaluation has focused on systems with up to 100 cells and with limited direct biological applications. To apply NeuralCPM to large-scale biological tissues, for example in cancer research, we identify three directions for future work. The first is accelerating the NeuralCPM metropolis kinetics, which pose computational challenges for large systems. We hypothesize that the use of efficient sampling techniques for discrete EBMs may alleviate this challenge (Grathwohl et al., 2021; Zhang et al., 2022; Sun et al., 2023). The second limitation concerns the global receptive field of the Neural Hamiltonian architecture. For the applications we considered, a global receptive field is biologically plausible, but this is not the case for tissue-scale simulations. To this end, we need to develop a Neural Hamiltonian in which each cell can only sense its immediate surroundings. The third limitation is the assumption of dynamics towards an equilibrium distribution. While this assumption is well-motivated in scenarios like embryo development, other applications concern actively migrating cells, leading to non-Markovian dynamics. A conditional Hamiltonian that also depends on the system's history can address this limitation. In addition, another promising research direction is to use NeuralCPM to discover biological mechanisms, for example by using explainable AI techniques for neural network models of dynamical systems (Cranmer et al., 2020; Brunton et al., 2016). Finally, to foster adoption of NeuralCPM by biologists and integrate with other phenomena, e.g. cell division, NeuralCPM can be incorporated into widely-used software packages for CPM simulation (Starruß et al., 2014; Swat et al., 2012).

## Acknowledgements

We thank Satoshi Toda for sharing the microscopy data on bipolar axial self-organization. We are also grateful to the organizers of the workshop 'Simulating tissue dynamics with cellular Potts models' for facilitating the meeting and discussions that led to this project. We furthermore thank Quirine J. S. Braat from Eindhoven University of Technology for her comments on the cellular Potts related sections of the manuscript. K.M. acknowledges that this work used the Dutch national e-infrastructure with the support of the SURF Cooperative using grant no. EINF-7724. L.B. acknowledges support by the German BMBF (grants 031L0293D, 031L0315A).

## Impact Statement

Modeling multicellular dynamics is crucial for biology, and accurate computer simulations can contribute to accelerated research on a wide range of biological phenomena. However, it remains imperative to validate the simulation results against experiments, especially when relying on neural simulation models. We do not foresee any potential negative societal consequences other than those associated with general research in machine learning and cell biology.

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

## A. Data Generation

**Cell sorting**    Training data for fitting an analytical cell sorting Hamiltonian was generated by sampling states with the energy function given in equation (4), where in this case $H_{\text{case-specific}} = 0$. Inspired by the original proposal of the CPM by Graner & Glazier (1992), we consider cells of two distinct cell types that perform cell sorting due to differential adhesion. We distinguish the scenarios a, b, d, and f from Edelstein-Keshet & Xiao (2023), characterized by different contact energies between cells, which are laid out in Tables 4 to 7. The Lagrange multiplier was set to $\lambda_V = 0.1$ for scenario a, $\lambda_V = 0.5$ for scenario b, $\lambda_V = 0.1$ for scenario d, and $\lambda_V = 0.05$ for scenario f. All sets of simulations were performed with a target volume of $V^* = 60$ and temperature $T = 1$ on a $100 \times 100$ grid with 50 cells, 25 of each type, which were initialized as single pixels randomly scattered within a centered circle with a radius of 25 lattice sites. The resulting datasets comprised 128 independent full lattice snapshots each.

*Table 4.* Contact energies for scenario a

| Medium | 0.0 | | |
|--------|-----|-----|-----|
| Type 1 | 0.5 | 0.333333 | |
| Type 2 | 0.5 | 0.2 | 0.266667 |
| | Medium | Type 1 | Type 2 |

*Table 5.* Contact energies for scenario b

| Medium | 0.0 | | |
|--------|-----|-----|-----|
| Type 1 | 2.5 | 1.0 | |
| Type 2 | 1.0 | 4.5 | 1.0 |
| | Medium | Type 1 | Type 2 |

*Table 6.* Contact energies for scenario d

| Medium | 0.0 | | |
|--------|-----|-----|-----|
| Type 1 | 15 | 4.0 | |
| Type 2 | 7.5 | 7.5 | 4.0 |
| | Medium | Type 1 | Type 2 |

*Table 7.* Contact energies for scenario f

| Medium | 0.0 | | |
|--------|-----|-----|-----|
| Type 1 | 2.0 | 7.0 | |
| Type 2 | 8.0 | 5.5 | 3.0 |
| | Medium | Type 1 | Type 2 |

**Cellular MNIST**    Ground truth data for the synthetic structural assembly experiment in section 4.3 was generated in a similar way to the cell sorting data described above. Notably, $H_{\text{case-specific}}(x)$ now took the form of an external potential

$$H_{\text{case-specific}}(x) = \sum_{i \in L} \mu(x_i)\phi_i \tag{8}$$

where $\mu(x_i)$ can be considered the coupling strength to the potential $\phi_i$. The coupling $\mu(x_i)$ was then defined so that only cells of type 2 couple to $\phi_i$ with constant strength $\bar{\mu} = 10$, while $\phi_i$ was chosen such that cells of type 2 favor arrangements that mimic handwritten digits from the MNIST data set (Deng, 2012). To that end, the MNIST images were binarized by applying a threshold at half brightness. Subsequently, a Euclidean distance transform was performed. Finally, the image was scaled with cubic interpolation from the input resolution $28 \times 28$ to the chosen domain size of $100 \times 100$ pixels. The result was used as $\phi_i$ in the Hamiltonian. The distance transform yields a sloped potential which pushes the type 2 cells into the shape of the desired digit. Differential adhesion similar to the cell sorting case was imposed to better separate cells of different types; see table 8. In addition to these parameters, we set $\lambda_V \approx 0.974$, $V^* = 100$, and $T = 1$ and initialized the system with the same procedure as above. The final data set contained 1280 samples.

*Table 8.* Contact energies for cellular MNIST

| Medium | 0.0 | | |
|--------|-----|-----|-----|
| Type 1 | 6.0 | 3.0 | |
| Type 2 | 6.0 | 6.0 | 3.0 |
| | Medium | Type 1 | Type 2 |

**Bipolar axial organization**    The experimental cell aggregates of Toda et al. (2018) consist of 200 to 240 cells in 3D, which amounts to about 8 cells along a diameter and about 40 cells in the cross-section. To generate the synthetic training data, we therefore consider 40 interacting cells, 20 of each type. We use Morpheus (Starruß et al., 2014) to randomly initialize a

cluster of these 40 cells, after which we assign each cell of type two a preferred motion in the direction of one of the two poles. This results in artificially creating configurations where each cell of type 2 has clustered together in the pole it was assigned to move to. In addition, the standard cell sorting Hamiltonian applied, with $V^*(c) = 150$, $\lambda = 1$, and contact energies as shown in Table 9. In addition, we set Morpheus' temperature parameter for this experiment to $T = 2.0$. Using this procedure, we generated 1000 samples, which we randomly rotate for training.

*Table 9.* Contact energies for synthetic bipolar axial organization

| Medium | 0.0 | | |
|--------|------|------|------|
| Type 1 | 16.0 | 6.0 | |
| Type 2 | 16.0 | 16.0 | 6.0 |
| | Medium | Type 1 | Type 2 |

## B. Implementation Details

Our implementation is built on JAX (Bradbury et al., 2018) and Equinox (Kidger & Garcia, 2021).

### B.1. Training and Sampling

**Training loop**  A pseudocode description of the training loop is given in Algorithm 1. We initialize the MCMC chains from datapoints. We use persistent chains instead of reinitializing in each iteration, following the Persistent Contrastive Divergence algorithm from (Tieleman, 2008), but with the regularization term from (Du & Mordatch, 2019). While using an approximation of the gradient of the max likelihood objective, this approach reduces the amount of compute per optimization step since less MCMC steps and thus less forward passes of $H_\theta(x)$ have to be performed. Note that the autodifferentiation step does not backpropagate through the sampling chain. Common hyperparameters are given in Table 10. We used the Adam

---

**Algorithm 1** NeuralCPM training procedure

---

**Input:** dataset $\mathcal{D} = \{x_n\}_{n=1}^N \overset{i.i.d}{\sim} p^*(x)$, learning rate $\eta$, number of sampling steps $K$, number of parallel flips $P$, sampler reset probability $p$
Initialize $K$ sampling chains $\{x_b^-\}_{b=1}^B \sim \mathcal{D}$
Initialize model $\theta$
**while** not converged **do**
    $\{x_b^+\}_{b=1}^B \sim \mathcal{D}$
    For each $x_b^-$, with probability $p$ reinitialize $x_b^- \sim \mathcal{D}$
    $\{x_b^-\}_{b=1}^B \leftarrow \text{ApproxPCPM}(H_\theta, K, P, \{x_b^-\}_{b=1}^B)$
    $\mathbf{g} \leftarrow \text{autodiff}(\hat{L}(\theta))$ (eq. 6)
    $\theta \leftarrow \text{Adam}(\eta, \theta, \mathbf{g})$
**end while**
return $\theta$

---

optimizer in all experiments with learning rate $\eta = 1e-3$ and standard hyperparameters $\beta_1 = 0.9, \beta_2 = 0.999, \epsilon = 1e-8$. The number of sampling steps per model update $K$ is determined in units of 'Monte Carlo Steps', i.e. the total size of the lattice $|L|$ (aka the grid size), as is common in the CPM literature. Thus, $K = |L| * \text{Monte Carlo steps}$. Each training run was terminated after 24 hours if it was not yet finished. If models diverged during training or started showing clear artifacts, we manually found the last model state that was not problematic and used this model state for the experiments. Finally, for the Cellular MNIST and bipolar axial organization experiments, we additionally assigned random types to each cell when the state of the system was reinitialized (second line within the while loop of Algorithm 1) to ensure a broader coverage of the state-space during training.

**Approximate sampler**  Estimating the loss function using an MCMC sampler that mixes quickly is imperative for scalable training, since each MCMC step requires a forward pass of $H_\theta(x)$ in the Metropolis-Hastings (MH) correction step. The original CPM sampler uses a proposal distribution that perturbs only one lattice site $l \in L$ at a time, resulting in a slow mixing rate and thus a very expensive training loop (Graner & Glazier, 1992). We instead use approximate parallelized CPM dynamics with a proposal distribution that samples multiple sites and changes their cell state in parallel.

*Table 10.* Training hyperparameters used in results of their respective section

| Hyperparameter | 4.2 | 4.3 | 4.4 |
|---|---|---|---|
| Batch size $B$ | 16 | 16 | 16 |
| Max. num training steps | 1e4 | 1e4 | 1e4 |
| Monte Carlo steps | 1.0 | 0.5 | 0.7 |
| Lattice size | 200x200 | 100x100 | 149x149 |
| EWA $\alpha$ | 0.0 | 0.99 | 0.99 |
| Regularizer $\lambda$ | 0.0 | 0.0005 | 0.0005 |
| Num parallel flips $P$ | 100 | 50 | 50 |
| Sampler reset probability $p_{reset}$ | 100% | 2.5% | 2.5% |
| Randomize cell types after reset | No | Yes | Yes |

---

**Algorithm 2** ApproxPCPM

---

**Input:** Hamiltonian $H_\theta$, number of sampling steps $\tau$, number of parallel flips $P$, initial state $x^0$
**for** $t$ in 1 to $\tau$ **do**
    $x' \leftarrow x^{t-1}$
    $\mathcal{B}^t \leftarrow \{l \in L | \exists j \in \mathcal{N}(l) : x_l^t \neq x_j^t\}$
    $\mathcal{S}^t = \{l_p\}_{p=1}^P \overset{i.i.d}{\sim} \mathcal{B}^t$
    **for** lattice site $i \in \mathcal{S}^t$ in parallel **do**
        $\mathcal{M}(i) = \{j \in \mathcal{N}(i) | x_i^t \neq x_j^t\}$
        $x_i' \leftarrow x_{j \sim \mathcal{M}(i)}^{t-1}$
        $\Delta \leftarrow H_\theta(x') - H_\theta(x^{t-1})$
        $p_i \leftarrow \min(1, e^{-\Delta})$
        $u_i \sim U(0,1)$
        $x_i^t \leftarrow x_i'$ if $u_i \leq p_i$
    **end for**
**end for**
Return $x^\tau$

---

Pseudocode for the approximate CPM dynamics is given in Algorithm 2. Let $\mathcal{N}(l)$ define the set of all neighboring lattice sites of lattice site $l$. The sampler uses a proposal distribution where first, $P$ lattice sites are independently and uniformly sampled from the **boundary** of cells. This boundary subset $\mathcal{B}$ contains all $l \in L$ that have a neighboring lattice site that belongs to a different cell than the cell at the lattice site in question. Such structure in the proposal causes state updates where the system (and system energy) actually changes, increasing the convergence speed of the sampler towards more meaningful configurations. After sampling a set of lattice sites $\mathcal{S}$ from the boundary $\mathcal{B}$, we sample for each site $i \in \mathcal{S}$ a lattice site that is mapped to a different cell in $C$, a set we denote as $\mathcal{M}(i) = \{j \in \mathcal{N}(i) | x_i^t \neq x_j^t\}$. Then, we copy that neighbouring cell into the originally sampled lattice site and **perform an MH correction step for each site in $\mathcal{S}$ in parallel**. That is, we perform $P$ MH correction steps in parallel on states where only one lattice site has been changed. Then, we keep all the proposed changes that were accepted and combine them together into the next system state.

The resulting transition probabilities do not satisfy detailed balance because we essentially use a faulty MH correction step, and thus we cannot guarantee that the system has a stationary distribution defined by $H_\theta(x)$. However, the standard CPM also does not satisfy detailed balance (Durand & Guesnet, 2016), and we found in preliminary experiments that this custom approximate sampler achieved speedups over even state-of-the-art discrete MCMC samplers such as (Grathwohl et al., 2021; Zhang et al., 2022; Sun et al., 2023). We believe this is because the custom proposal is able to leverage structure (the boundary constraint) in its proposal that is specific to the cellular dynamics problem, and that the gradient approximations of these discrete systems as used in state-of-the-art samplers were rather poor, likely due to the very unsmooth nature of the neural Hamiltonian $H_\theta(x)$.

## B.2. Model Details and Hyperparameters

Here, we discuss details on the (hyper)parameters of the various Hamiltonian models used in our experiments.

### B.2.1. ANALYTICAL HAMILTONIANS

The cell sorting Hamiltonian is exactly as described in Equation 4, where the parameters $J(c_1, c_2)$ and $\lambda$ are learnable; we set $V^*(c)$ to the average volume of all cells observed in the training data for all $c \in C$. The cell sorting model with external potential is the same as the cell sorting Hamiltonian, with the addition of an external potential as described in Equation 8. However, rather than setting the potential up-front as done in the data generation (Appendix A), we learn it through stochastic gradient descent.

### B.2.2. NEURAL HAMILTONIANS

**Initial embedding layer.** We first compress the sparse one-hot encoded representation to a dense representation by applying a learned downsampling through a single linear strided convolutional layer that operates on each cell independently. Both the stride and kernel size of this layer are $3 \times 3$ and $5 \times 5$ for Cellular MNIST and bipolar axial sorting respectively.

**NH layers.** For simplicity, we choose a fixed architecture for $\phi^l$ and $\psi^l$ throughout this work: both are two repetitions of $\{\text{Conv2D} \to \sigma\}$, where Conv2D is a convolution with a kernel size of 3 and $\sigma$ is the SiLU activation function (Elfwing et al., 2018). We use summation as permutation-invariant aggregation function $\bigoplus$. The hidden dimension (amount of channels) per cell for each NH layer increases progressively with the depth of the Neural Hamiltonian model, where the first Conv2D layer in both $\psi^l$ and $\phi^l$ maps the input to an output with `out channels` equal to this hidden dimension, and where subsequent Conv2D layers preserve the amount of channels within each NH layer:

$$\{h_c^l \in \mathbb{R}^{\texttt{in channels} \times h \times w}\} \to \phi^l \to \{h_c' \in \mathbb{R}^{\texttt{out channels} \times h \times w}\} \tag{9}$$

$$\{h_c^l \in \mathbb{R}^{\texttt{in channels} \times h \times w}, A \in \mathbb{R}^{\texttt{out channels} \times h \times w}\} \to \psi^l \to \{o_c \in \mathbb{R}^{\texttt{out channels} \times h \times w}\} \tag{10}$$

Additionally, we use a residual connection (He et al., 2016) for each NH layer, connecting its input $h_c^l$ with its output $o_c$ before max-pooling.

The specific hidden dimensions and pooling design for the Neural Hamiltonians is then as follows:

- Cellular MNIST: 4 NH layers

    - Hidden dimensions: [8, 16, 32, 32]
    - Max-pooling downsampling rates: [3, 2, 1, 1]

- bipolar axial sorting: 6 NH layers

    - Hidden dimensions: [16, 32, 32, 64, 64, 64]
    - Max-pooling downsampling rates: [2, 1, 2, 1, 2, 1]

We then apply a linear convolution with 32 output channels, before pooling over all cells and pixels using summation to get a vector representation of the system that is invariant to both permutations and translations. This is then processed by an MLP with two hidden layers with SiLU nonlinearity and 32 neurons each before mapping to a scalar output with a single linear layer. Similar to the NH layers, the MLP layers model the residual with respect to the input.

For the 1-layer Neural Hamiltonian baseline, only the first of these NH layer is applied before pooling over the representation of all cells using pixel-wise summation.

**NH-based baseline models.** The Neural Hamiltonian+closure model uses the neural network to model an additive term on top of the cell sorting Hamiltonian (Equation 4), where we take the same approach as in Section B.2.1 for the analytical component.

The 1-layer NH + CNN model uses the first layer of the respective Neural Hamiltonians as described earlier in this section, before pooling over all cells but not over all pixels. This yields a grid representation of the system that is invariant to permutations, which is subsequently processed by a CNN architecture consisting of blocks of two {Conv2D → $\sigma$} repetitions, and a residual connection between the input and output of each block. The first layer of the convolution block maps the input to the specified hidden dimension, which remains the same for the second layer. We again apply max-pooling to the output of each block to get a compressed representation of the system. The specific designs are as follows:

- Cellular MNIST: 3 convolution blocks
  - Hidden dimensions: [32, 64, 128]
  - Max-pooling downsampling rates: [1, 2, 2]

- bipolar axial sorting: 4 convolution blocks
  - Hidden dimensions: [32, 64, 64, 128]
  - Max-pooling downsampling rates: [1, 2, 1, 2]

The output is then processed in the exact same way as the output of the NH layers described in the paragraph above, with the exception that we already pooled over all cells before the CNN, and thus only aggregate over all pixels.

## C. Additional Results

### C.1. Fitting Analytical Hamiltonians

The convergence plot of the parameters for type A, type D, and type F cell sorting (analogous to Figure 4, see Edelstein-Keshet & Xiao (2023)) can be found in Figure 8. As in Figure 4, the parameters converge rapidly to their target values over the course of training, demonstrating robustness of the training process over different parameterizations of the Hamiltonian. Additionally, Table 11 shows the mean and standard deviation of the fitted parameter values calculated over five independent initializations of the model parameters. The results show that the parameters robustly converge to a value close to the target, regardless of their initialization.

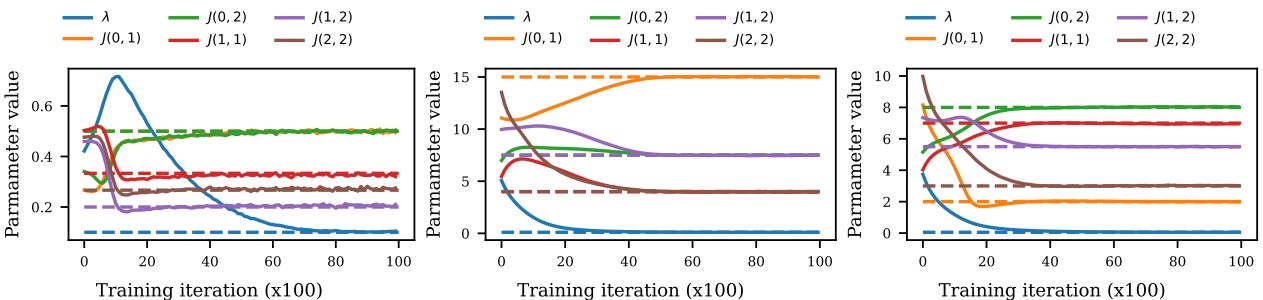

*Figure 8.* Convergence of the parameters for cell sorting type A (left), type D (center), and type F (right). Dashed lines indicate the true values, solid lines indicate the learned values ($T = T^*$) over the course of training. The parameters converge rapidly to the true values for all parameterizations of the cell sorting Hamiltonian.

### C.2. Additional Results for Baselines and Ablations

In addition to the results presented in Sections 4.3 and 4.4, we consider three additional baselines and ablations for the architectural design of the Neural Hamiltonian. The first is a GNN Hamiltonian, which enjoys permutation symmetry but lacks translation symmetry and inductive biases for lattice data. The second is an ablation of the Neural Hamiltonian where cells cannot interact with each other, achieved by removing the cell-interaction CNN module from each layer and masking out any information from neighboring cells. Finally, we consider an ablation in which we remove the local pooling step in each layer of the Neural Hamiltonian. Qualitative and quantitative results are shown in Figure 9 and Table 12, respectively.

As expected, we see that the Neural Hamiltonian without cell interactions fails to guide cells towards realistic self-organized states. This is because each cell needs to know the position of other cells in order to move towards a location that contributes

*Table 11.* Additional results for the parameter fitting experiment. 'Mean' and 'Std.' columns refer to the mean and standard deviation of the fitted values of the respective parameter, calculated over five independent random parameter initializations.

| Parameter | Type A | | | Type B | | | Type D | | | Type F | | |
|---|---|---|---|---|---|---|---|---|---|---|---|---|
| | True value (reference) | Mean | Std. | True value (reference) | Mean | Std. | True value (reference) | Mean | Std. | True value (reference) | Mean | Std. |
| $\lambda$ | 0.1 | 0.103 | 0.001 | 0.5 | 0.616 | 0.002 | 0.1 | 0.101 | 0.000 | 0.05 | 0.051 | 0.000 |
| $J(0,1)$ | 0.5 | 0.499 | 0.003 | 2.5 | 2.441 | 0.008 | 15.0 | 15.022 | 0.005 | 2.0 | 1.995 | 0.009 |
| $J(0,2)$ | 0.5 | 0.497 | 0.006 | 1.0 | 1.012 | 0.007 | 7.5 | 7.504 | 0.020 | 8.0 | 8.039 | 0.014 |
| $J(1,1)$ | 0.333 | 0.327 | 0.005 | 1.0 | 1.056 | 0.003 | 4.0 | 3.951 | 0.006 | 7.0 | 6.947 | 0.011 |
| $J(1,2)$ | 0.2 | 0.207 | 0.003 | 4.5 | 4.494 | 0.003 | 7.5 | 7.478 | 0.021 | 5.5 | 5.502 | 0.015 |
| $J(2,2)$ | 0.267 | 0.274 | 0.007 | 1.0 | 1.048 | 0.006 | 4.0 | 4.004 | 0.011 | 3.0 | 3.021 | 0.007 |

to a realistic cellular structure. The GNN-based Hamiltonian results in structures that somewhat resemble the cellular structures of the data, but with substantially lower quality than the models based on our Neural Hamiltonian architecture. The Neural Hamiltonian without any pooling yields good results on cellular MNIST, but diverged during training for bipolar axial sorting (the remaining hyperparameters were identical to NH+closure), and hence we do not report these data. A possible reason for this could be the higher dimensionality of the lattice. Finally, all reported models performed well from an individual cell perspective, suggesting that the biology-informed term is generally effective.

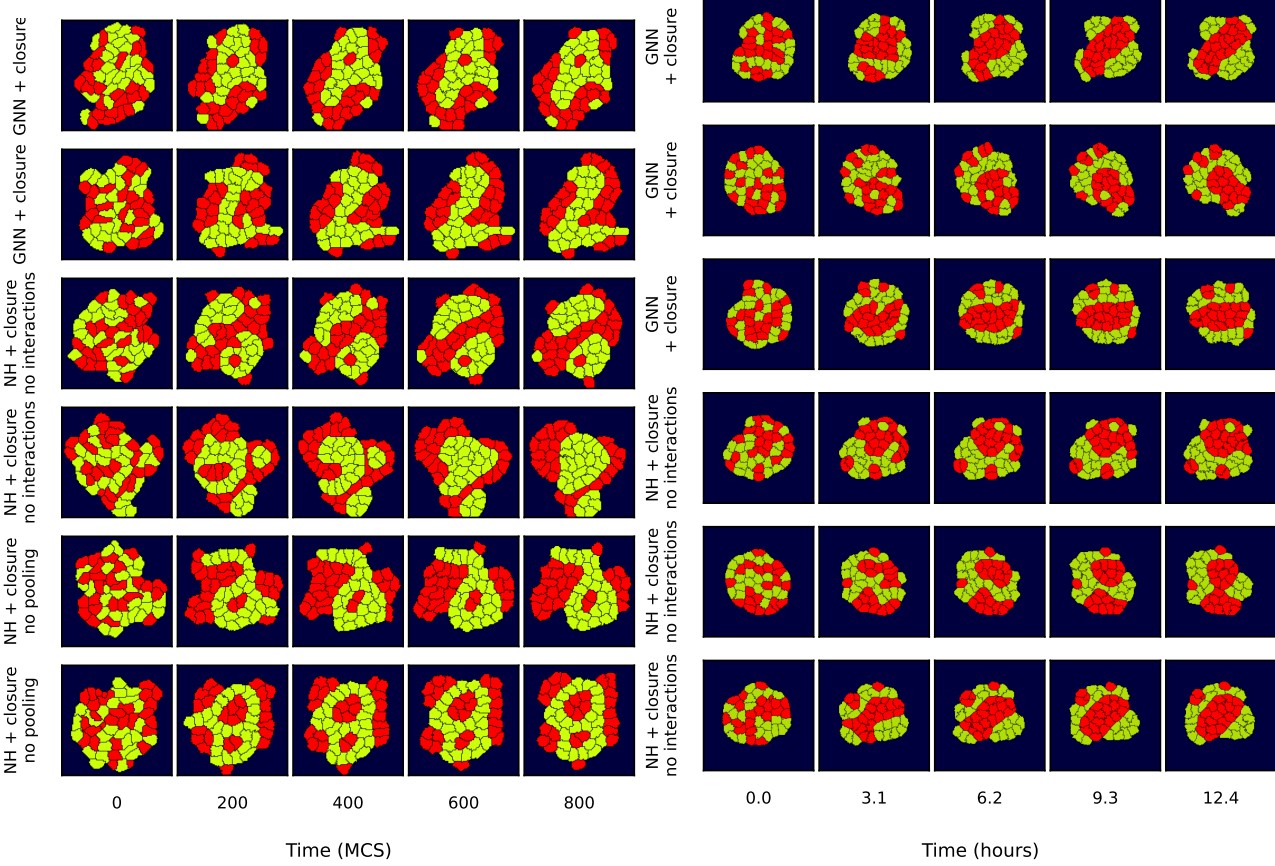

*Figure 9.* Qualitative results for additional baselines and ablations on Cellular MNIST (left) and bipolar axial sorting (right).

To stress the importance of using a Neural Hamiltonian for the bipolar axial organization experiment, Figure 10 shows the development of bipolar organization over time, similar to Figure 7(b), but now for a CPM with cell sorting Hamiltonian. For type-2 cells, there is reasonable overlap, implying that the degree of collinearity along the principal axis of type-2 cells is similar to the NeuralCPM case. However, the type-1 cells do not align, meaning that the analytical model fails to 'squeeze'

*Table 12.* Quantitative results of additional baselines and ablations for Cellular MNIST and bipolar axial organization experiments.

| Model/ablation | Cellular MNIST | | | bipolar axial organization | | |
|---|---|---|---|---|---|---|
| | $p_{\text{volume}}$ ↑ | $p_{\text{unfragmented}}$ ↑ | CS ↑ | $p_{\text{volume}}$ ↑ | $p_{\text{unfragmented}}$ ↑ | Axial alignment RMSE ↓ |
| GNN + closure | **1.00** | **1.00** | 3.15 | **1.00** | **1.00** | 100.4 |
| Neural Hamiltonian No interactions + closure | **1.00** | 0.97 | 3.06 | **1.00** | **1.00** | 119.3 |
| Neural Hamiltonian No pooling + closure | **1.00** | **1.00** | 4.24 | N/A | N/A | N/A |
| Neural Hamiltonian + closure (cq. Tables 2 and 3) | **1.00** | **1.00** | **4.35** | 0.77 | **1.00** | **37.3** |

them along the orthogonal axis, which NeuralCPM successfully learned.

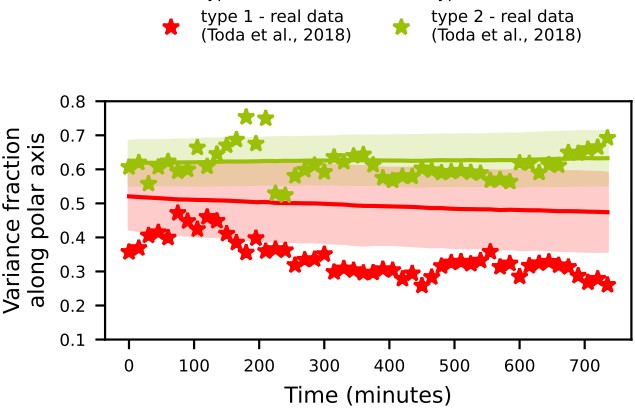

*Figure 10.* Development of bipolar cellular organization over time for a CPM with Cellsort Hamiltonian.

## C.3. Additional Qualitative Results

Additional qualitative simulation results for the Cellular MNIST and bipolar axial organization experiments can be found in Figures 11 and 12 respectively.

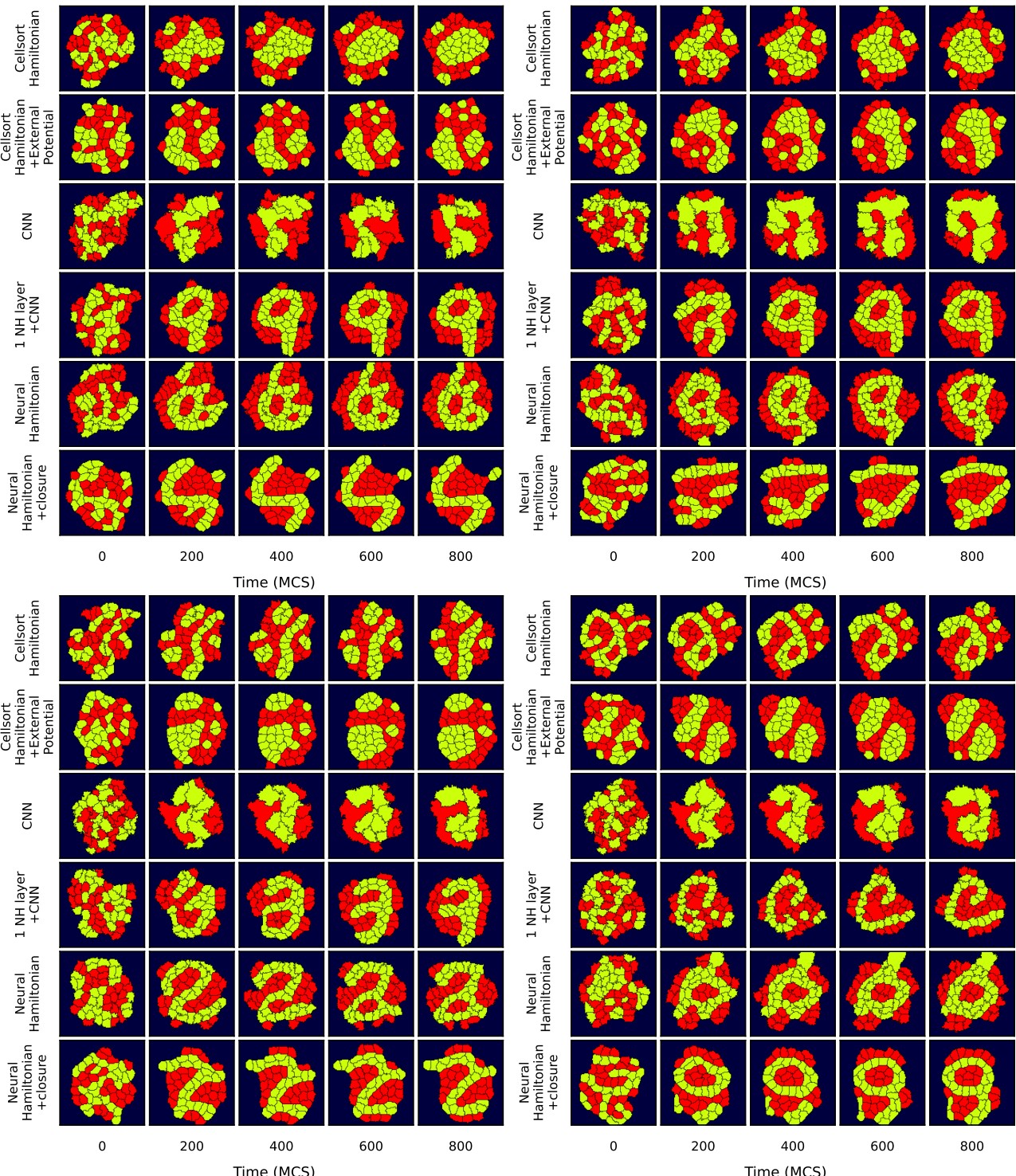

*Figure 11.* Additional qualitative results for Cellular MNIST simulations.

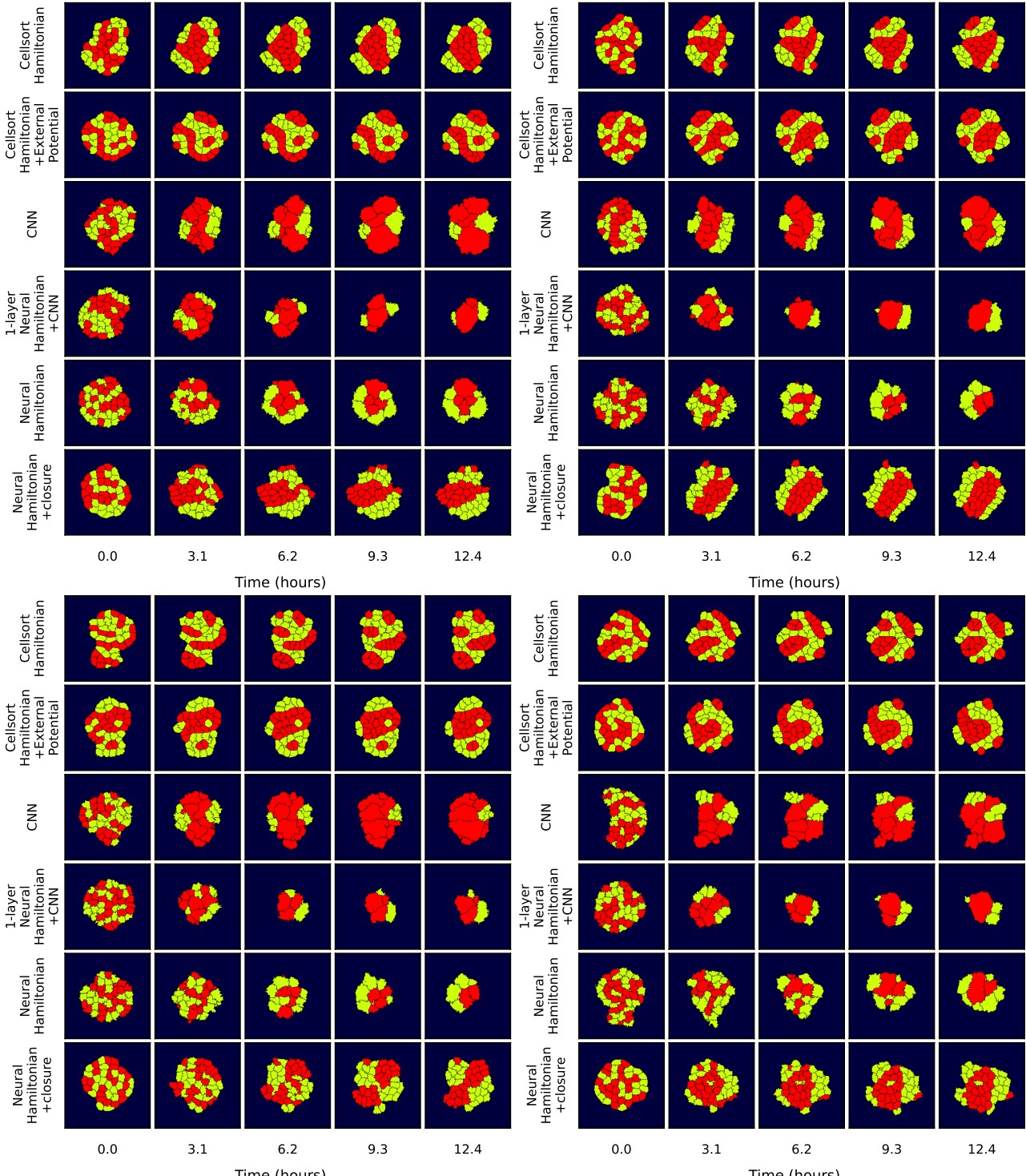

*Figure 12.* Additional qualitative results for bipolar axial sorting simulations.

