# OpenReview forum: "Deep Neural Cellular Potts Models"
_ICML.cc/2025/Conference — ICML 2025 poster_

### Official Review · Reviewer_U1hT · 2025-02-22

**Overall Recommendation:** 5

**Summary:**

The paper introduces NeuralCPM, a novel cellular Potts model (CPM) that employs a neural network to parameterize the Hamiltonian, diverging from traditional CPMs that rely on manually-defined, physics-inspired analytical Hamiltonians.

This Neural Hamiltonian is designed to respect symmetries inherent in cellular dynamics, such as permutation and translation invariance, and can be trained directly on observational data. A key feature is its ability to integrate domain knowledge through a hybrid model, combining known biological mechanisms with the neural network to enhance biological realism and expressiveness.

The authors evaluate NeuralCPM across three scenarios: synthetic data for parameter fitting of known Hamiltonians, the Cellular MNIST dataset to demonstrate increased expressiveness in modeling complex structures, and real-world bi-polar axial organization data from Toda et al. (2018) to showcase practical applicability.

Main findings include NeuralCPM's superior performance over traditional CPMs in capturing complex cellular dynamics, achieving higher biological consistency and pattern fidelity, as evidenced by metrics like the Classifier Score (CS) and axial alignment RMSE.

## update after rebuttal

The authors have clarified my concerns. I think this is a strong paper, as a result, I am increasing my score.

**Claims And Evidence:**

The primary claim is that NeuralCPM is more expressive and better at modeling complex cellular dynamics than traditional CPMs. This is supported by:

- Synthetic Data Experiments: NeuralCPM accurately fits parameters of known analytical Hamiltonians for cell sorting, validated by convergence plots (e.g., Figures 4 and 8), showing rapid alignment with ground-truth values.
- Cellular MNIST Experiment: The model generates digit-like cellular structures, with a higher CS (Table 1) than analytical baselines, indicating improved expressiveness. Qualitative results (Figure 5) further support this.
- Real-World Data Experiment: NeuralCPM predicts bi-polar axial organization, aligning with laboratory observations (Figure 7), with a significantly lower axial alignment RMSE (37.3) compared to baselines (Table 2).

The evidence is generally clear and convincing.

**Essential References Not Discussed:**

I don't think there is related work that is essential but not discussed in this paper.

**Experimental Designs Or Analyses:**

The authors use three different experiments. Overall, the designs are solid, with minor concerns addressed via questions below.

**Methods And Evaluation Criteria:**

The overall approach of learning the Hamiltonian that was previously manually defined analytically makes sense.

The Neural Hamiltonian architecture is thoughtfully designed, using convolutional neural networks (CNNs) and permutation-invariant aggregation to respect cellular dynamics symmetries, which is apt for the problem.

The hybrid model integrating domain knowledge is a sensible approach, enhancing interpretability and leveraging biological insights.

The evaluation seems appropriate. However, for real-world data, direct comparison metrics between simulated and observed dynamics could further validate the approach.

**Other Comments Or Suggestions:**

- Typos: "classs" (Page 2, line 109) should be "class"; "desings" (Page 14) should be "designs".
- Formatting: Ensure consistent figure captions (e.g., Figure 7a vs. 7(a)).

**Other Strengths And Weaknesses:**

### Strengths

- Originality: Combining neural networks with CPMs is innovative and useful, removing the need for manual Hamiltonian design.
- Significance: Improved modeling of complex biological systems could impact fields like cancer research.
- Domain Knowledge Integration: The hybrid model enhances trust and interpretability for biologists.
- Clarity: The paper is very well written and very clear.

### Weaknesses

- Computational Complexity: MCMC sampling limits scalability, though future work is proposed.
- Training Stability: Challenges with deep EBM training are noted but not fully resolved, affecting reproducibility.

**Questions For Authors:**

Synthetic Data for Bi-Polar Experiment: Could you elaborate on how the synthetic training data for the bi-polar axial organization experiment was generated to reflect real-world dynamics? Specifically, how were cell motion preferences assigned, and what ensures their biological plausibility?

**Relation To Broader Scientific Literature:**

NeuralCPM advances CPM research by integrating neural networks.

**Theoretical Claims:**

This paper does not introduce new theoretical results.

---

> ### Author Rebuttal · Authors · 2025-03-31
>
> Thank you for the rigorous review. We are happy to learn that you appreciated the originality, significance and clarity of our work.
> We address your specific comments in our response below;
> any new tables and figures (referred to as e.g. Figure R1) can be found via https://anonymous.4open.science/r/neuralcpm-rebuttal-results-761F.
>
> ### General comments
>
> 1. **Direct comparison between simulated and observed dynamics:**
> In Figures 6 and 7b of the paper, we show a qualitative respectively quantitative comparison of the simulated dynamics and the observed real-world biological cell dynamics of Toda et al.
> Both figures show that the simulations of NeuralCPM align with the observations.
> To further emphasize this, we now include a new Figure R4, following the same procedure as Figure 7b, but now using simulations based on the Cellsort Hamiltonian.
> Comparing Figure 7b and R4, we see that NeuralCPM’s dynamics align substantially better with the observations of Toda et al.
>
>
> ### Weaknesses
>
> 2. **Computational complexity:**
> Indeed, MCMC sampling limits scalability to some extent, but we would like to stress that our current applications did not suffer from prohibitive computation times.
> To illustrate, the amortized time for a single simulation with the Neural Hamiltonian model was 44 seconds for cellular MNIST (Figure 5) and 62 seconds for bi-polar axial organization (Figure 6), and all models were trained for at most 24 hours.
> Additionally, as we briefly mentioned in the conclusion of our paper, we expect that exploiting a first order Taylor expansion to approximate the energy difference [1,2] can accelerate the computation substantially, since this can enable approximating the energy difference for many parallel spin flips at the cost of a single Hamiltonian evaluation. Moreover, recent advances in Deep EBM training (e.g. [3]) can also be applied to NeuralCPM in the future.
>
> 3. **Training stability:**
> NeuralCPM relies on EBM training techniques, which can be sensitive to hyperparameters.
> However, for our experimental scenarios, we found that adding the biological term to the Hamiltonian greatly stabilized training.
> This is because the biological term effectively acts as a ‘guardrail’ that constrains the MCMC simulation to a biologically reasonable manifold (e.g. cells have reasonable volumes and are contiguous objects).
> We further expect that ongoing research on training (discrete) EBMs (e.g. [1,2,3,4] as well as future works) can carry over to NeuralCPM training.
>
> ### Questions
>
> 4. **Clarification on synthetic data for bi-polar experiment:**
> To generate synthetic training data corresponding to Toda et al., we used cell motion along imposed trajectories such that cells of one type (‘red’) move from their random initial position into a central band and cells of the other type leave the central band towards opposing cluster borders.
> So by construction, we generate training data with a bi-polar axis.
> Additionally, to make sure the training data are isotropic, we randomly rotate the final configuration.
> In contrast, biological cells (and NeuralCPM) do not rely on any such prescribed trajectories, but update cell positions as a result of cell-cell interactions in the current configuration.
> However, the patterns that result from this procedure are comparable to those observed in Toda et al. (2018) in terms of bi-polar axial organization (see Fig. 7).
> Finally, our model needs only these final patterns for training, and a comparison of the model with the real observations of Toda et al. shows that the spontaneous symmetry-breaking dynamics of NeuralCPM align with the dynamics observed in Toda et al. (2018).
>
> ### Other comments or suggestions
>
> 5. **Typos and formatting:**
> We thank the reviewer for their detailed examination of the manuscript, and we will make sure to remove these errors in the updated version of the paper.
>
> ### References
>
> [1] Grathwohl et al. (2021). Oops I Took A Gradient: Scalable Sampling for Discrete Distributions.
>
> [2] Zhang et al. (2022). A Langevin-like sampler for discrete distributions.
>
> [3] Schröder et al. (2023). Energy discrepancies: a score-independent loss for energy-based models.
>
> [4] Sun et al. (2023). Discrete Langevin samplers via Wasserstein gradient flow.

---

### Official Review · Reviewer_ZLYN · 2025-03-14

**Overall Recommendation:** 3

**Summary:**

In this thesis, we propose NeuralCPM, a neural network-based cellular Potts model (CPM), which aims to learn the dynamics of multicellular systems through neural Hamiltonian. The core idea of the method is to use Deep Energy-Based Model (EBM) to train Hamiltonian in order to overcome the problem of traditional CPM which relies on hand-designed energy functions. The paper claims that NeuralCPM is able to fit more complex cell behaviors and demonstrates its superiority by comparing it with experimental data.
The paper provides several experiments to support these claims, including:
1. a synthetic data experiment to verify that NeuralCPM can learn the parameterized Hamiltonian and accurately simulate cellular MNIST data.
2. Cellular MNIST dataset experiments: demonstrate that NeuralCPM can generate morphology-specific cellular arrangements, such as number shapes.
3. real biological experiments (Toda et al., 2018): to validate the effectiveness of NeuralCPM in the bipolar axial organization task.

**Claims And Evidence:**

1. The core methodology proposed in the paper, NeuralCPM and its Neural Hamiltonian, is conceptually clear and reasonable, especially the emphasis on the symmetry of the cellular system, the integration of domain knowledge, and the rational use of the deep energy model (EBM) framework, which is supported by sufficient theoretical basis, and the argumentation process is logical, rigorous, and clear, and the proposed theories and methodologies are credible and innovative in their own right. The proposed theories and methods are credible and innovative.
2. In terms of experiments, the validity and advantages of the method are verified through three different scenarios (parameter fitting, complex structure simulation, and real biological data application). The experiments are well-designed, and the quantitative metrics (e.g., RMSE, Classifier Score, etc.) can clearly reflect the performance of the model. The experimental results clearly show that the Neural Hamiltonian model outperforms the traditional analytical Hamiltonian model and the basic neural network architecture in simulating the complex cellular behaviors. However, the article mentions that “the CNN-based Hamiltonian produces unrealistic dynamics in the absence of substitution symmetry”, which suggests that the limitations of the chosen baseline approach are more obvious and may affect the comprehensiveness of the experimental comparisons.
3. despite the overall strong evidence, there are still some aspects of the paper that need further refinement. For example, the authors rely on synthetic data as the training set in the real biological data validation experiments, and the sample size of the real experiments is on the low side (only six repetitions of the experiments), and whether the resulting generalization ability is sufficient for generalization to other real systems still needs to be further verified. In addition, the stability of the training process still needs to be verified and illustrated in different scenarios.

**Essential References Not Discussed:**

None

**Experimental Designs Or Analyses:**

I thoroughly checked the soundness and validity of the experimental designs and analyses presented in the manuscript. The paper primarily conducts three sets of experiments: fitting known analytical Hamiltonians, simulating complex multicellular structures using Cellular MNIST data, and applying the model to a real-world biological dataset (bi-polar axial organization).

The experimental design for parameter fitting using analytical Hamiltonians is sound and well-structured. The authors employed Root Mean Squared Error (RMSE) to quantitatively assess the accuracy of parameter estimation, clearly demonstrating the model’s capability in learning physically meaningful parameters from data. However, given the central role of the temperature parameter in CPM dynamics, the experimental setup could benefit from further clarification on temperature sensitivity and robustness.

In the Cellular MNIST and bi-polar axial organization experiments, the authors selected appropriate metrics, such as cell volume, fragmentation, and Classifier Score, to capture both biological realism and structural complexity effectively. These metrics convincingly evaluate both cellular-level and collective-level model performance. However, there are some concerns: firstly, the choice of baseline models is limited—particularly the CNN baseline lacks permutation symmetry, leading to unrealistic dynamics. Including more suitable baselines or justifying these choices explicitly would strengthen the comparative validity of results. Secondly, in the bi-polar axial organization scenario, synthetic training data may oversimplify biological complexity, and the potential gap between synthetic and real-world data could be more explicitly discussed. Addressing these concerns would enhance the robustness and generalizability of the experimental validation.

**Methods And Evaluation Criteria:**

1. The authors have selected reasonable evaluation metrics in their experiments, including biological consistency metrics for cell volume and fragmentation, and Classifier Score for quantifying the structural expressiveness of cell populations, and this evaluation clearly reflects the performance of the model. In addition, the authors use several experimental scenarios with different difficulties, including parameter fitting of theoretical Hamiltonian functions, complex image structures (Cellular MNIST) and real biological experiments (e.g., bipolar cellular organization experiments by Toda et al.), to demonstrate the generality and validity of the model, and in particular, the comparative validation with real experimental data enhances the credibility of the results.

2. However, some of the experiments in the paper have certain limitations, such as the simulation data constructed simplifies the complexity of real biological systems to a certain extent (e.g., there are discrepancies between the synthesized data and the real data in the bi-polar axial organization experiments), and may not be able to fully reflect the nonlinear and stochastic characteristics of the real cellular dynamics scenarios. In addition, there is a clear lack of selection of CNN baseline models, which makes it possible that some of the resultant comparisons may underestimate the capabilities of other potential architectures.

**Other Comments Or Suggestions:**

None

**Other Strengths And Weaknesses:**

None

**Questions For Authors:**

Q1. In the bi-polar axial organization experiment, synthetic data generated by Morpheus were used for training. Could you elaborate on how well these synthetic data represent real-world complexity? Specifically, how sensitive is your model to the simplifications made during data generation? Clarifying this would strengthen confidence in your method’s applicability to real biological scenarios.

Q2. You introduced an approximate sampler to accelerate the MCMC dynamics. Could you explain more explicitly how deviations from detailed balance affect your results? Have you empirically or theoretically evaluated how these deviations impact the final distributions obtained by your trained models? Addressing this point would help assess potential biases introduced by the approximation.

Q3. Baselines such as the CNN-based Hamiltonian lacked permutation symmetry, potentially limiting their comparative value. Could you justify the selection of these baseline models? Alternatively, did you consider other relevant baselines that respect the symmetries in cellular systems? This clarification is important to fully understand the relative advantages of your proposed architecture.

**Relation To Broader Scientific Literature:**

None

**Theoretical Claims:**

This paper does not contain an explicit proof of theory, so there is no need to check the correctness of the proof. The main contribution of the authors is to propose a deep learning method called NeuralCPM and to clarify the soundness of the theoretical framework by means of experiments and a symmetry constrained exposition.

---

> ### Author Rebuttal · Authors · 2025-03-31
>
> Thank you for the thorough review! We appreciate that you found our methodology clear, innovative, and rigorous.
> We address your specific comments in our response below; any new tables and figures (referenced as e.g. Figure R1)
> can be found via https://anonymous.4open.science/r/neuralcpm-rebuttal-results-761F.
>
> ### General comments
>
> 1. **Temperature sensitivity:**
> Since we can always rescale the Hamiltonian in the CPM, we can assume without loss of generality that the temperature equals 1, and train the Hamiltonian for that temperature.
> The typical effect of $T$ roughly corresponds to the strength of the fluctuations in the system over time [1];
> however, it does not substantially affect the behavior of the system at equilibrium, and a particular Hamiltonian will result in qualitatively similar self-organized patterns for varying temperatures.
>
> ### Questions
>
> 2. **Use of synthetic data for bi-polar axial organization experiment:**
> To generate synthetic training data corresponding to Toda et al., we used cell motion along imposed trajectories such that cells of one type (‘red’) move from their random initial position into a
> central band and cells of the other type leave the central band towards opposing cluster borders.
> So by construction, we generate training data with a bi-polar axis.
> Additionally, to make sure the training data are isotropic, we randomly rotate the final configuration.
> Biological cells (and NeuralCPM) do not rely on any such prescribed trajectories, but update cell positions as a result of cell-cell interactions in the current configuration.
> Still, we argue that the synthetic final self-organized states are statistically comparable to those observed in Toda et al. (2018) in terms of bi-polar organization, shown in Figure 7a.
> As the training process only relies on these final states, the simplifications we made in the data generation do not necessarily cause adverse effects to the model quality.
> We test this in Figure 7, where NeuralCPM’s simulations are shown to align very well with the real data observations of Toda et al.
> Of course, in general one should take such a synthetic approach with care, and in this light this experiment should be seen as a proof-of-principle towards more complex biological applications.
> We will update our paper to include an explicit discussion about this point.
>
> 3. **Approximate parallel CPM sampler and deviations from detailed balance:**
> Notably, the simulation dynamics from the standard CPM already lack detailed balance [2].
> Though the approximate parallel sampler can theoretically suffer from synchronization issues when parallel spin-flips affect each other (see e.g. [3]), we stress that it should be regarded as a surrogate sampler to accelerate training;
> other established training techniques for EBMs, like Persistent Contrastive Divergence [4], also successfully sacrifice unbiased sampling for accelerated training, as unbiased maximum likelihood estimation is generally intractable.
>
> 4. **Lack of permutation symmetry in CNN baseline:**
> Based on your suggestion, we now added an additional GNN baseline which enjoys permutation symmetry to further strengthen our experimental evaluation, in addition to other ablations (see point 1 in our response to reviewer KQ4z).
> The quantitative and qualitative results are shown in Table R2 and Figures R2+R3 respectively.
> The results indicate that the Neural Hamiltonian outperforms the GNN baseline in terms of self-organization (CS and alignment RMSE metrics), emphasizing the importance of inductive biases for lattice data, e.g. translation symmetry and locality, in addition to permutation symmetry.
>
> ### References
>
> [1] Glazier and Graner (1993). Simulation of the differential adhesion driven rearrangement of biological cells.
>
> [2] Durand and Guesnet (2016). An efficient Cellular Potts Model algorithm that forbids cell fragmentation.
>
> [3] Sultan et al. (2023). A parallelized cellular Potts model that enables simulations at tissue scale.
>
> [4] Tieleman (2008). Training restricted Boltzmann machines using approximations to the likelihood gradient.

---

### Official Review · Reviewer_KQ4z · 2025-03-23

**Overall Recommendation:** 4

**Summary:**

This paper introduces Neural Hamiltonians, a novel approach for parameterizing the Hamiltonian function in cellular Potts models. The authors propose using neural networks to learn the Hamiltonian function directly from data, while preserving important physical and biological constraints.

The key innovation is a formulation that respects critical symmetries inherent to cell populations - specifically, invariance to cell permutations and lattice translations. This ensures the learned models maintain biologically plausible behavior regardless of how cells are indexed or positioned.

Additionally, the authors present a hybrid framework that combines traditional symbolic Hamiltonians (which incorporate known biological principles) with neural network-based Hamiltonians (which can capture complex patterns from data). The approach includes learnable weights that balance the contribution of these two components, allowing the model to leverage both domain knowledge and data-driven insights.

## update after rebuttal

After examining the authors' rebuttal, I can confirm that they have adequately addressed my concerns. Therefore, I have upgraded my recommendation to Accept.

**Claims And Evidence:**

See the strengths and weaknesses below.

**Essential References Not Discussed:**

N/A

**Experimental Designs Or Analyses:**

The experimental methodology and analytical approach used in this work are fundamentally sound and well implemented. The authors have come up with a reasonable framework for evaluating their Neural Hamiltonian method. While the overall experimental design is sound, there are several specific observations and suggestions regarding these experiments that I have detailed in the sections below.

**Methods And Evaluation Criteria:**

See the strengths and weaknesses below.

**Other Comments Or Suggestions:**

The abbreviation CNN is used once in the text before using the full term.

**Other Strengths And Weaknesses:**

**Strengths**

- The paper is well-written, with clear explanations of complex concepts that make the work accessible despite the interdisciplinary nature of the research.

- The authors' approach of using neural networks to parameterize Hamiltonian functions in cellular Potts models represents an interesting innovation. This technique appears to be both effective and mathematically sound.

- The Neural Hamiltonian framework successfully captures complex non-local cell interactions that traditional analytical methods struggle to model, at least as could be observed in the cellular MNIST experiments. This capability can be a substantial advancement in the field of computational biology and multicellular modeling.

- The combination of symbolic (biologically-informed) and neural network-based Hamiltonians is particularly valuable. This can be an elegant balance between incorporating established biological knowledge and leveraging data-driven insights.

**Weaknesses**

- The baseline models included in the experimental comparisons are somewhat simplistic. For instance, the CNN baseline is predictably ill-suited for the task, making its poor performance unsurprising.

- The connection between the synthetic datasets (particularly cellular MNIST) and real cellular dynamics remains vague. The paper does not sufficiently justify how success on these artificial benchmarks translates to biological validity or relevance.

- The experiments with real biological data are notably limited in scope and depth. These experiments fail to fully demonstrate the potential advantages of Neural Hamiltonians in addressing meaningful biological questions. More extensive validation with diverse real-world cellular data would strengthen the paper's claims and better highlight the practical utility of the method.

- The paper would benefit from more detailed ablation studies to isolate and quantify the contribution of each component of the proposed framework, particularly regarding the hybrid approach and symmetry constraints.

**Questions For Authors:**

- How well does the cellular MNIST dataset reflect the complexity and characteristics of real cell population dynamics? Could you elaborate on the specific biological phenomena that this synthetic benchmark captures, and what limitations it might have in representing actual cellular behaviors?

- Could you provide the visualization referenced in Figure 7b for the cellsort Hamiltonian? This would help me better understand the comparative performance of your method in this specific scenario.

- For the first experiment, which components of the Neural Hamiltonian design are necessary for accurate parameter estimation? Have you conducted ablation studies to determine how removing or modifying certain elements affects the overall performance?

- Given the novelty of your architecture, it would be valuable to see a systematic ablation study on the architectural components and their impact on experimental outcomes. Specifically: How crucial is the cell interaction CNN to the overall performance? What is the specific contribution of the pooling layers in capturing cellular dynamics? How does removing symmetry constraints affect model performance? What is the optimal balance between the symbolic and neural components in the hybrid approach?

- How does the computational complexity of training and inference with Neural Hamiltonians compare to traditional approaches? Is there a significant trade-off between model expressivity and computational requirements?

**Relation To Broader Scientific Literature:**

See the summary.

**Theoretical Claims:**

N/A

---

> ### Author Rebuttal · Authors · 2025-03-31
>
> Thank you for your insightful review. We are glad that you found our work well-written and sound, and that you appreciated the innovations of our approach.
> Our response can be found below; new results (referenced as e.g. Figure R1)
> can be found via https://anonymous.4open.science/r/neuralcpm-rebuttal-results-761F.
>
> ### Weaknesses
>
> 1. **Baselines and ablations:**
> We now added three additional baselines and ablations: 1)
> a GNN Hamiltonian, which enjoys permutation symmetry but lacks translation symmetry and inductive biases for lattice data; 2)
> an ablation of the Neural Hamiltonian where cells cannot interact by removing the cell-interaction CNN in each layer, masking out any information from neighboring cells; and 3)
> an ablation of the Neural Hamiltonian where we remove the local pooling step in all NH layers.
> We show the results for experiments 2 and 3 in Table R2 and Figures R2 and R3.
> For the bi-polar axial organization experiment, the ablation of removing the pooling in the NH layers diverged during training (remaining hyperparameters identical to NH+closure), and hence we do not report these data.
> The overall results show that the paper's Neural Hamiltonian + closure design consistently outperforms all baselines in terms of the converged multicellular patterns.
>
> 2. **Biological relevance of synthetic data:**
> We agree that the cellular MNIST data is not directly biologically relevant.
> Instead, for this experiment we aimed to design a scenario with cellular structures that are complex enough to be poorly approximated by a symbolic model,
> yet simple enough to facilitate straightforward (visual) validation of model quality.
> In a broader biological context, the relevance of this experiment is to test NeuralCPM’s ability to model structures beyond the capabilities of traditional CPMs, which indicates our method's potential to bring real biological use cases such as wound healing, cell migration, and embryo formation within reach of CPM simulations.
>
> 3. **Scope of biological experiments:**
> We only evaluated one real biological use-case in this paper, as our goal was to introduce and validate the core concepts of NeuralCPM before moving to more extensive and complex biological case studies.
> Still, we argue that the results on this scenario sufficiently demonstrate NeuralCPM’s potential for
> future applications, as the CPM formalism is very versatile (evidenced by hundreds of CPM models of biological processes in public repositories: https://artistoo.net/examples.html, https://morpheus.gitlab.io/#examples, https://compucell3d.org/Simulation%20Movies),
> and can now be derived from data with NeuralCPM.
>
> 4. **Ablation study:**
> Please see point 1 above.
>
> ### Questions
>
> 5. **MNIST dataset:**
> Please see point 2 above.
>
> 6. **Visualization for Cellsort Hamiltonian:**
> Please see Figure R4.
> For type-2 cells, there is reasonable overlap, implying that
> the degree of collinearity along the principal axis of type-2 cells is similar to the NeuralCPM case.
> However, the type-1 cells do not align,
> meaning that the Cellsort model fails to ‘squeeze’ them along the orthogonal axis, which NeuralCPM successfully learned.
>
> 7. **Hamiltonian components:**
> The goal of the experiment is to investigate if the NeuralCPM training framework can recover unknown parameters in symbolic Hamiltonians.
> To this end, the Hamiltonian consisted only of symbolic terms and randomly initialized parameters.
> We now also added additional results to investigate the robustness of this approach;
> please see point 1 of our response to reviewer FiGo for a discussion.
>
> 8. **Ablation study:**
> Please see point 1 above.
>
> 9. **Computational complexity:**
> Due to the Neural Hamiltonian, NeuralCPM requires more computational effort than traditional CPMs with symbolic Hamiltonians.
> The amortized time for a simulation with the Neural Hamiltonian model was 44 seconds (Figure 5) and 62 seconds (Figure 6), while the Cellsort Hamiltonian + External potential, which exploits locality assumptions for an efficient calculation, required 0.3 and 0.5 seconds respectively.
> So, although the Neural Hamiltonian is more expensive, the simulation times remain manageable.
> Further, approximating the energy difference with a first order Taylor expansion [1,2] can further accelerate the computation since this enables approximating the energy difference for multiple spin flips at the cost of a single Hamiltonian evaluation.
> So, using such a technique to evaluate 10+ spin flips in parallel could give an order of magnitude speedup, offering a promising path for scaling NeuralCPM to tissue-scale simulations.
> Finally, all training runs took at most 24 hours, and the tradeoff in terms of relative time per training step is comparable to inference.
>
> ### References
>
> [1] Grathwohl et al. (2021). Oops I Took A Gradient: Scalable Sampling for Discrete Distributions.
>
> [2] Zhang et al. (2022). A Langevin-like sampler for discrete distributions.

---

### Official Review · Reviewer_FiGo · 2025-03-25

**Overall Recommendation:** 3

**Summary:**

This paper introduces NeuralCPM, a neural network-based approach to cellular Potts modeling (CPM) for simulating collective cell dynamics. Traditional CPMs rely on physics-inspired Hamiltonians that require significant domain expertise to design and may not fully capture complex biological behaviors. NeuralCPM parameterizes the Hamiltonian with a neural network architecture specifically designed to respect symmetries in cellular dynamics, allowing it to be trained directly from observational data. The framework also enables the integration of known biological mechanisms by combining analytical Hamiltonians with the neural network in a hybrid model. The authors validate their approach through experiments on synthetic data and real-world multicellular systems, demonstrating that NeuralCPM can model cellular dynamics that traditional analytical Hamiltonians cannot capture.

**Claims And Evidence:**

The first experiment convincingly demonstrates that NeuralCPM can accurately recover parameters from cell sorting models, validating the learning algorithm. The second experiment with the Cellular MNIST dataset shows that NeuralCPM can capture complex spatial arrangements that analytical Hamiltonians cannot, with quantitative metrics supporting this claim. The final experiment applying the approach to real biological data demonstrates practical utility.

**Essential References Not Discussed:**

No.

**Experimental Designs Or Analyses:**

For the parameter recovery experiment (Section 4.2), the methodology is valid but could be more rigorous. The authors correctly generate synthetic data using known parameters and evaluate RMSE between learned and ground truth values. However, the rapid convergence shown in Figure 4 suggests this might be a simplified test case. The experiment would be more convincing with diverse initial conditions and parameter regimes.

In the Cellular MNIST experiment (Section 4.3), the classifier score metric is innovative but potentially problematic - success at recreating visually recognizable digits may not indicate biologically meaningful dynamics. The complementary biological metrics (volume and fragmentation) help address this concern, making the overall analysis approach sound.

For the bi-polar organization experiment with real biological data (Section 4.4), the axial alignment variance is an appropriate domain-relevant metric. However, the experimental design lacks statistical testing across multiple simulation runs to establish the significance of the reported improvements. Additionally, while the training instability issue with pure neural models is honestly reported, more analysis of this limitation would strengthen the paper. Additionally, including neural competing methods could further strengthen the validity of this experiment.

**Methods And Evaluation Criteria:**

The experimental methodology is sound, with appropriate metrics to evaluate both biological realism (cell volume, fragmentation) and the ability to capture complex collective behaviors (classifier score, axial alignment). The authors properly acknowledge limitations and discuss hyperparameter sensitivity.

**Other Comments Or Suggestions:**

No

**Other Strengths And Weaknesses:**

**Strengths:**

- The Neural Hamiltonian architecture is thoughtfully designed to respect fundamental symmetries (permutation, translation) in cellular systems
- The hybrid approach combining analytical and neural components is particularly valuable, improving both biological realism and training stability
- The method requires only observations of self-organized states for training, not full trajectories
- The validation against real biological data demonstrates practical utility beyond synthetic examples

**Weaknesses:**

- The approximate sampling compromises theoretical foundations
- Architecture design choices lack sufficient justification and ablation studies
- Limited to equilibrium dynamics, excluding many biological processes
- No clear pathway for scaling to larger cellular systems
- Training energy-based models can be unstable, as evidenced by the divergence of some models in the bi-polar sorting experiment

**Questions For Authors:**

- How computationally expensive is NeuralCPM compared to traditional CPMs, both for training and inference? This information would be valuable for potential users considering adopting this approach.

- How sensitive is the model to the choice of temperature parameter T in the CPM? Since you noted this is "poorly identifiable from static snapshots," have you explored whether NeuralCPM could learn an appropriate effective temperature automatically?

- For the bi-polar axial organization experiment, you mention that Neural Hamiltonian models without the closure term failed due to "fast divergence during training." Could you elaborate on specific techniques you tried to stabilize these models before resorting to the hybrid approach?

- Does the training instability of pure neural Hamiltonians indicate a fundamental limitation of the approach? What modifications might address this?

- What other neural competing methods might be applicable to the microscopy time-lapse data by Toda et al. (2018)?

**Relation To Broader Scientific Literature:**

The paper positions NeuralCPM within the intersection of energy-based models from machine learning and cellular Potts models from computational biology. The authors correctly identify the gap in the literature - while neural networks have been used to accelerate physical simulations, their application to improving the expressivity of cellular dynamics models is novel.

**Theoretical Claims:**

The paper contains few formal theoretical claims requiring proof verification. I checked the mathematical formulation of the permutation invariant neural Hamiltonian architecture in Section 3.1, which appears correct in its construction using permutation-invariant aggregation functions.

---

> ### Author Rebuttal · Authors · 2025-03-31
>
> Thank you for the thorough review and valuable suggestions.
> We are glad to hear that you found our Neural Hamiltonian well-designed and that you appreciated the validation against real biological data.
> We address all comments and questions of your review below;
> new tables and figures (referred to as e.g. Table R1) can be found through https://anonymous.4open.science/r/neuralcpm-rebuttal-results-761F.
>
> Due to the character constraint, we refer to responses to other reviewers where indicated.
>
> ### General comments
>
> 1. **More initial conditions and parameter ranges for parameter recovery experiment:**
> We now also tested the approach on more parameter regimes, based on Type D and Type F cell sorting of [1].
> Further, we tested five different random initializations of the parameters of the Hamiltonian to assess the robustness of the method.
> The results, shown in Table R1 and Figure R1, demonstrate that the parameters robustly converge to values close to the ground-truth for all four scenarios.
>
> 2. **Statistical testing for bi-polar axial organization:**
> We now added a statistical analysis of the NH + closure model compared to all baseline models as follows:
> we take 5 groups of 10 simulation runs, calculate the axial alignment RMSE as in Table 2 for each group, and then test whether the mean error of the NH + closure model is significantly lower than the respective baseline.
> We used Welch’s t-test, and applied a Bonferroni correction to control for multiple testing.
> The results, shown in Table R3, indicate that the mean error is significantly lower than all baseline models.
>
> ### Weaknesses
>
> 3. **Approximate parallel sampling:**
> Please see point 3 of our response to reviewer ZLYN.
>
> 4. **Architecture design choices, ablation studies, additional baselines:**
> Please see point 1 of our response to reviewer KQ4z for a discussion on the newly added baselines and ablations.
>
> 5. **Going beyond equilibrium dynamics:**
> NeuralCPM can be extended by also conditioning the Hamiltonian on earlier observed states, and training on full trajectory data.
> This can be interpreted as learning the energy difference, instead of the absolute value.
> We already conducted preliminary experiments with this approach that gave promising initial results,
> but they are not yet ready for publication, and we leave this extension for future work.
>
> 6. **Scaling to larger systems:**
> Please see point 9 of our response to reviewer KQ4z.
>
> 7. **Training stability:**
> See point 10 below.
>
> ### Questions
>
> 8. **Computational requirements:**
> Please see point 9 of our response to reviewer KQ4z.
>
> 9. **Sensitivity to temperature $T$:**
> Since we can always rescale the Hamiltonian in the CPM, without loss of generality, we can assume that the temperature parameter $T$ equals 1, and train the Hamiltonian for that temperature.
> The typical use of $T$ roughly corresponds to a parameter that determines the strength of the fluctuations in the system over time [2];
> however, it does not substantially affect the behavior of the system at equilibrium.
>
> 10. **Techniques to stabilize training, and whether this is a fundamental limitation:**
> We applied established best practices from the literature for mitigating training instabilities when training discrete EBMs:
> We used Persistent Contrastive Divergence [3], similar to [4], and we regularized the Hamiltonians during training (Section 3.2).
> We also tried different architectural configurations during preliminary experiments, e.g. network depth.
> Although these techniques are relevant for training EBMs in general, adding the symbolic biology-informed term to the Neural Hamiltonian effectively mitigated training stability issues for our use-case.
> This is due to the biological terms functioning as ‘guardrails’ for the simulation, preventing biologically unrealistic behaviors.
> Finally, discrete EBMs are an active field of research, and recent advancements (e.g. [4,5,6]) are also applicable to NeuralCPM.
>
> 11. **Other neural competing methods:** To the best of our knowledge, there are no earlier proposed neural methods that apply to the cellular dynamics data of Toda et al. and rely only on equilibrium data.
>
> ### References
>
> [1] Edelstein-Keshet and Xiao (2023). Simplified Cell Sorting — Morpheus contributed examples.
> https://morpheus.gitlab.io/model/m2007.
>
> [2] Glazier and Graner (1993). Simulation of the differential adhesion driven rearrangement of biological cells.
>
> [3] Tieleman (2008). Training restricted Boltzmann machines using approximations to the likelihood gradient.
>
> [4] Grathwohl et al. (2021). Oops I Took A Gradient: Scalable Sampling for Discrete Distributions.
>
> [5] Zhang et al. (2022). A Langevin-like sampler for discrete distributions.
>
> [6] Schröder et al. (2023). Energy discrepancies: a score-independent loss for energy-based models.

---

### Decision · Program_Chairs · 2025-05-01

**Decision:**

Accept (poster)

**Comment:**

This paper introduces Neural Hamiltonians, a novel approach for parameterizing the Hamiltonian function in cellular Potts models. The reviewers agreed that the experiments strongly support the capability of the Neural Hamiltonians model in simulating collective cell dynamics. However, reviewers raised concerns regarding potential limitations in biologically realistic simulation experiments, insufficient ablation studies to justify architectural choices, and limited baseline comparisons. After examining the reviews and rebuttals, I'm convinced that the authors have adequately addressed most concerns. While there remains room for improvement in demonstrating the model's robustness, particularly in real biological applications, I believe this paper offers a promising first step in this direction. Therefore, I recommend acceptance.